# Podocyte OTUD5 alleviates diabetic kidney disease through deubiquitinating TAK1 and reducing podocyte inflammation and injury

Ying Zhao[1,2,5], Shijie Fan[2,5], Hong Zhu[1,5], Qingqing Zhao[2], Zimin Fang[3], Diyun Xu[3], Wante Lin[2,3], Liming Lin[1,2], Xiang Hu[1], Gaojun Wu[1], Julian Min[2] & Guang Liang ®[1,2,4] ✉

Recent studies have shown the crucial role of podocyte injury in the development of diabetic kidney disease (DKD). Deubiquitinating modification of proteins is widely involved in the occurrence and development of diseases. Here, we explore the role and regulating mechanism of a deubiquitinating enzyme, OTUD5, in podocyte injury and DKD. RNA-seq analysis indicates a significantly decreased expression of OTUD5 in HG/PA-stimulated podocytes. Podocyte-specific *Otud5* knockout exacerbates podocyte injury and DKD in both type 1 and type 2 diabetic mice. Furthermore, AVV9-mediated OTUD5 overexpression in podocytes shows a therapeutic effect against DKD. Mass spectrometry and co-immunoprecipitation experiments reveal an inflammation-regulating protein, TAK1, as the substrate of OTUD5 in podocytes. Mechanistically, OTUD5 deubiquitinates K63-linked TAK1 at the K158 site through its active site C224, which subsequently prevents the phosphorylation of TAK1 and reduces downstream inflammatory responses in podocytes. Our findings show an OTUD5-TAK1 axis in podocyte inflammation and injury and highlight the potential of OTUD5 as a promising therapeutic target for DKD.

Diabetic kidney disease (DKD), a prevalent chronic kidney disease, is the main cause of end-stage renal disease progression[1,2]. DKD is characterized by increased albuminuria, decreased glomerular filtration, thickening of the glomerular basement membrane (GBM), accumulation of mesangial matrix, and aggravation of podocyte injury[3–5]. Podocytes, the highly differentiated epithelial cells that line the outer surface of the GBM, are crucial for maintaining the integrity of the GBM[6,7]. The decrease in the number or density of podocytes leads to membrane destruction and promotes proteinuria[8]. Therefore, the degree of podocyte injury is usually used as an index to evaluate the progression of DKD[9]. In addition, the accumulation of inflammation in podocytes is associated with aggravated podocyte injury[10,11].

Identifying key proteins that regulate podocyte damage may provide strategies and targets for DKD therapy and hold great clinical significance.

As a post-translational modification, ubiquitination/deubiquitination plays a vital role in regulating protein stability, localization, and function, participating in the pathophysiology of various diseases[12]. Deubiquitinating enzymes (DUBs) negatively regulate the dynamic and reversible process of ubiquitination by removing ubiquitin molecules from the substrate proteins[13]. Current research on the regulation of renal function by DUBs primarily focuses on renal cancer and renal fibrosis. For instance, the deletion of USP11 in renal tubular epithelial cells improves renal function in mouse models with renal fibrosis by

[1]Department of Endocrinology, the First Affiliated Hospital of Wenzhou Medical University, Wenzhou, Zhejiang 325035, China. [2]Chemical Biology Research Center, School of Pharmaceutical Sciences, Wenzhou Medical University, Wenzhou, Zhejiang 325035, China. [3]Department of Cardiology, the First Affiliated Hospital of Wenzhou Medical University, Wenzhou, Zhejiang 325035, China. [4]School of Pharmaceutical Sciences, Hangzhou Medical College, Hangzhou, Zhejiang 310014, China. [5]These authors contributed equally: Ying Zhao, Shijie Fan, Hong Zhu. ✉e-mail: wzmcliangguang@163.com

promoting the degradation of EGFR[14]. OTUD1 and USP22 have been shown to promote fibrosis and injury in renal tubular epithelial cells[15,16]. In addition, USP13 promotes tumorigenesis of clear cell renal cell carcinoma through deubiquitinating ZHX2[17]. However, limited studies have explored the relationship between DUBs and podocyte injury in DKD. Considering the importance of podocyte function in DN, we performed an RNA sequencing (RNA-seq) analysis using podocytes under the high-concentration glucose and palmitic acid (HG/PA) condition and found a potentially DKD-related DUB in podocytes, ovarian tumor deubiquitinase 5 (OTUD5).

OTUD5 is a cysteine protease belonging to the OTU protein family[18]. It contains two highly conserved domains: the catalytic OTU domain and the ubiquitin-interacting motif (UIM) domain[19]. Recent studies have demonstrated the crucial role of OTUD5 in mediating innate immunity and inflammation development[18,20]. Furthermore, the role of OTUD5 in various cancers has been gradually discovered, such as in hepatocellular carcinoma and non-small cell lung cancer, where OTUD5 plays a carcinogenic role by removing the ubiquitination of TRIM25 and tumor protein P53[21]. However, to our knowledge, the role of OTUD5 in podocytes remains unknown.

In this study, we investigated the impact of OTUD5 on podocyte injury in diabetic mouse models. Mechanically, we found that OTUD5 interacts with TGF-β-activated kinase 1 (TAK1) and inhibits its phosphorylating activation by removing K63-linked ubiquitin from TAK1, thereby inactivating TAK1-MAPK pro-inflammatory signaling pathway in HG/PA-challenged podocytes. Overall, we revealed a OTUD5-TAK1 axis in podocyte injury and identified OTUD5 as a potential target for the treatment of DKD.

## Results

### Identification of OTUD5 as a regulator of podocyte injury
To identify DUBs associated with podocyte injury in DKD, we performed RNA-seq on the mouse podocyte cell line MPC5 treated with HG/PA. Gene set enrichment analysis (GSEA) revealed that HG/PA treatment activated the inflammatory signaling pathways, indicating successful induction of podocyte inflammation and injury (Supplementary Fig. 1a). Subsequently, we analyzed the DUB gene changing profile in these podocytes and found seven differentially expressed DUB genes, among which *A20* and *Otud5* showed the most significant downregulation (Fig. 1a, b). Since the functional role of A20 in podocytes has been previously reported[22], we focus on the OTUD5 in this study. We performed qPCR analyses and confirmed the downregulated mRNA level of *Otud5* in HG/PA-induced podocytes (Fig. 1c). The protein expression of OTUD5 was also decreased by HG/PA stimulation in a time-dependent manner in podocytes (Fig. 1d, Supplementary Fig. 1b). To reveal the clinical relevance of OTUD5, we collected renal biopsies from diabetic subjects and nondiabetic control subjects and examined the OTUD5 levels in these biopsies by immunofluorescence staining. As shown in Fig. 1e, the levels of OTUD5 in the glomerulus were decreased in diabetic subjects compared to nondiabetic subjects. In vivo, the levels of OTUD5 protein and mRNA in the renal cortex of mice with either type 2 diabetes mellitus (T2DM) (Fig. 1f, Supplementary Fig. 1c) or type 1 diabetes mellitus (T1DM) (Fig. 1g, Supplementary Fig. 1c) were also decreased compared with their control counterparts. We also examined OTUD5 levels in the kidney tissues from NOD and db/db diabetic mice, two well-established mouse models of spontaneous diabetes independent of STZ. Similar results were observed in the renal cortex of NOD and db/db mice (Fig. 1h, i, Supplementary Fig. 1d). We further isolated primary podocytes from T2DM or T1DM mice and observed a decrease in OTUD5 expression similarly (Fig. 1j, k).

Next, we examined the regulatory effect of OTUD5 on podocyte injury. The overexpression of OTUD5 in MPC5 podocytes (Supplementary Fig. 1e) significantly suppressed the increases in *Il6* and *Tnfa* mRNA levels induced by HG/PA (Fig. 1l). Moreover, overexpression of

OTUD5 attenuated podocyte apoptosis (Fig. 1m) and recovered the expression levels of Nephrin in podocytes (Fig. 1n) with HG/PA treatment. We also validated these results by overexpressing OTUD5 in OTUD5-knockdown MPC5 podocytes (Supplementary Fig. 1f–j). These results indicate OTUD5 as a negative regulator of podocyte injury in DKD.

### Podocyte-specific *Otud5* knockout aggravates podocyte injury and DKD in T2DM mice
To elucidate the potential role of OTUD5 in podocyte injury and DKD, we generated podocyte-specific *Otud5* knockout mice (conditional knockout, OTUD5CKO) by crossing *Otud5fl/fl* mice and *Nphs1-iCre* mice (Fig. 2a), which were identified by tail genotyping (Supplementary Fig. 2a). The deletion of OTUD5 in podocytes was validated by western blot analysis of isolated primary podocytes (Supplementary Fig. 2b). We used the high-fat diet feeding plus injection streptozotocin (HFD/STZ) method to induce T2DM[23,24] in both *Otud5fl/fl* mice and OTUD5CKO mice (Fig. 2b). Both HFD/STZ-treated groups displayed hyperglycemia after STZ treatment, and podocyte deficiency of OTUD5 did not affect the increase of blood glucose level induced by HFD/STZ (Fig. 2c). No significant differences in body weight change were found among these four groups (Supplementary Fig. 2c). However, OTUD5CKO mice exhibited more severe renal injury in T2DM mice, as indicated by serum creatinine, urea nitrogen, and urinary albumin/creatinine examinations (Fig. 2d–f). The degree of mesangial matrix expansion measured by H&E and PAS staining was increased in OTUD5CKO T2DM mice compared to *Otud5fl/fl* mice (Fig. 2g). Electron microscopy confirmed that OTUD5 deletion exacerbated T2DM-induced podocyte injury, including thickening of the GBM, and broadening and effacement of foot processes (Fig. 2h–k). Immunofluorescence staining of Nephrin further validated the enlarged degree of podocyte injury in OTUD5CKO-T2DM mice (Fig. 2l). Podocyte deficiency of OTUD5 also increased the mRNA levels of *Il6* and *Tnfa* in renal tissues of T2DM mice (Fig. 2m, n). Moreover, we observed that OTUD5CKO increased the chemokine levels (*Ccl2* and *Cxcl10*) in renal tissues of T2DM mice (Supplementary Fig. 2d, e), leading to more macrophage infiltration in diabetic kidneys (Supplementary Fig. 2f). Together, podocyte-specific *Otud5* knockout aggravates podocyte injury and DKD in T2DM mice.

### OTUD5CKO exacerbates podocyte injury and DKD in T1DM mice
We also explored the role of OTUD5 deficiency in mice with T1DM induced by STZ injection (Supplementary Fig. 3). Similarly, podocyte deficiency of OTUD5 did not affect the T1DM-induced increase in blood glucose levels (Fig. 3a). Examinations on serum creatinine, urea nitrogen, and urinary albumin/creatinine revealed that the renal functional injury was significantly aggravated by OTUD5CKO in T1DM mice (Fig. 3b–d). H&E staining and PAS staining followed similar trends in renal structural injury (Fig. 3e). Electron microscopy and Nephrin staining revealed that OTUD5CKO caused more severe podocyte injury in T1DM mice (Fig. 3f–j). Similarly, deficiency of OTUD5 in podocytes enhanced inflammatory *Il6* and *Tnfa* gene expression in T1DM mouse kidneys (Fig. 3k, l).

### Identification of TAK1 as a potential substrate protein of OTUD5
As a DUB, OTUD5 acts by deubiquitinating the substrate proteins. To identify the substrate of OTUD5 in podocytes, we carried out an LC-MS/MS analysis using OTUD5-overexpressing MPC5 cells (Fig. 4a). Among the top 15 OTUD5-binding proteins identified by the quantitative LC-MS/MS analysis (Supplementary Fig. 4a), TAK1 attracted our attention (Fig. 4b) as previous studies have demonstrated the pivotal regulatory role of TAK1 in podocyte inflammation and DKD progression[25,26]. Therefore, we hypothesize that OTUD5 negatively regulates inflammation and injury through deubiquitinating TAK1 in podocytes.

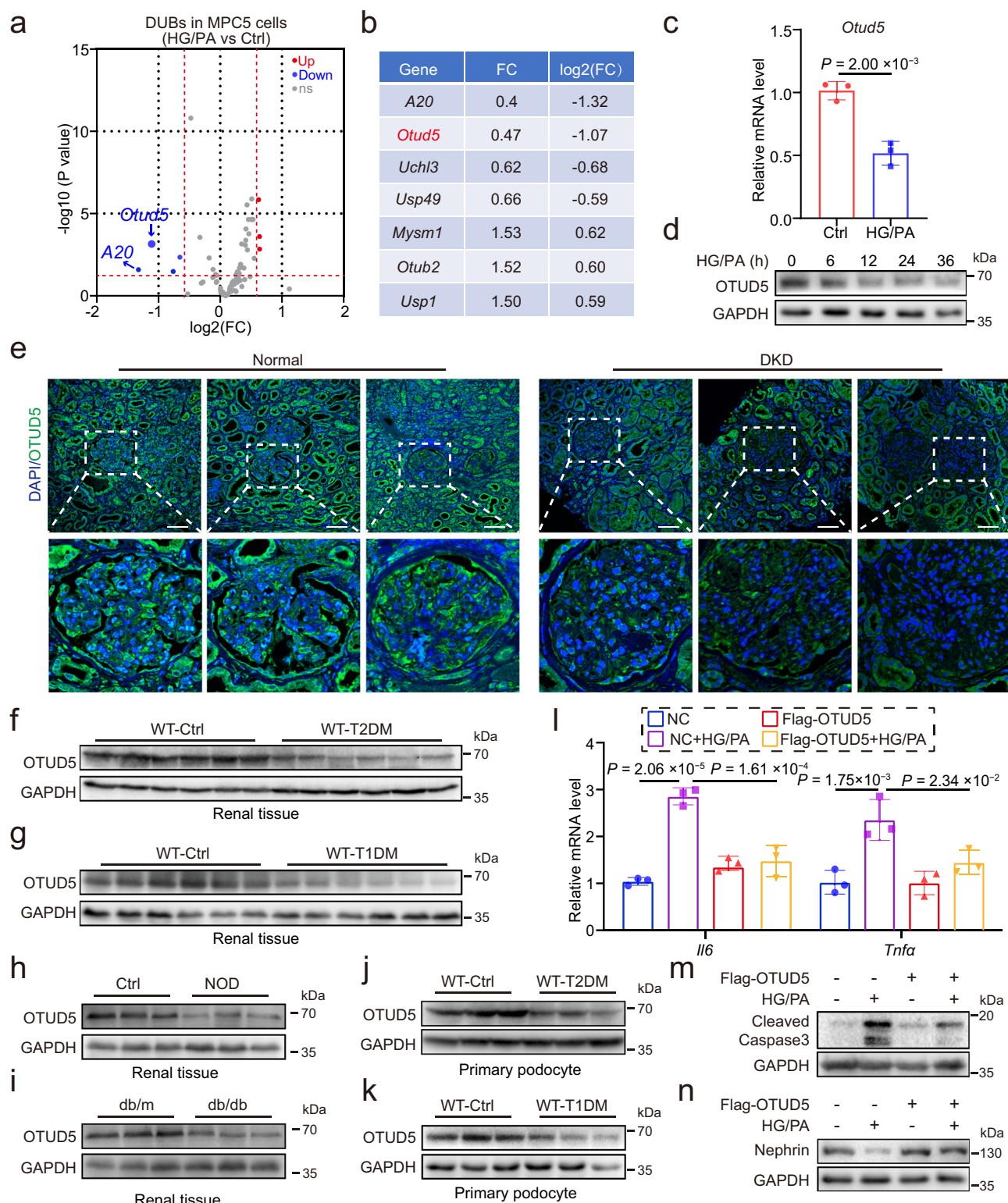

We confirmed the endogenous interaction between OTUD5 and TAK1 in podocytes using a Co-immunoprecipitation (co-IP) assay (Fig. 4c). Similarly, endogenous OTUD5 and TAK1 could form a complex in mouse kidney tissues (Fig. 4d). Moreover, Flag-OTUD5 and His-TAK1 plasmids were co-transfected into NIH/3T3 cells, confirming the exogenous interaction between OTUD5 and TAK1 (Fig. 4e). Next, we explored the mechanism of OTUD5 on TAK1 deubiquitination. As shown in Fig. 4f, OTUD5 overexpression decreased the polyubiquitination of TAK1 in NIH/3T3 cells. Further, we co-transfected TAK1 plasmid and K48- or K63-mutated ubiquitin plasmids in NIH/3T3 cells with or without OUTD5 and observed that OTUD5 reduced K63-linked ubiquitination of TAK1, rather than K48-linked ubiquitination (Fig. 4g). The Cysteine 224 residue (C224) in OTUD5 has been reported as the essential site for the deubiquitinating activity[20]. In contrast to wild-type OTUD5, the catalytically inactivated OTUD5 C224S mutant, which could interact with TAK1 normally (Supplementary Fig. 4b), failed to remove K63-linked ubiquitin molecules from TAK1 (Fig. 4h). These data revealed that OTUD5 cleaves the K63-linked ubiquitination of TAK1 via its active site C224.

**Fig. 1 | Identification of OTUD5 as a regulator of podocyte inflammation and injury. a** A volcano plot analysis illustrating the differential expression of DUBs induced by HG/PA in podocytes. ($n = 3$ samples for each Ctrl group and HG/PA group; $P$ values were determined by Wald test from DESeq2 software with Benjamini-Hochberg's correction). **b** A table shows DUBs with significant differences in MPC5 cells treated with HG/PA. **c** The mRNA level of *Otud5* in HG/PA-induced MPC5 cell lines. ($n = 3$ independent experiments; $P$ values were determined by two-tailed unpaired *t*-test and data are presented as mean ± SD). **d** Representative western blot of OTUD5 expression in MPC5 cell lines after stimulation with HG/PA for different durations. ($n = 3$ independent experiments). **e** Representative immuno-fluorescence (IF) images of OTUD5 expression in human renal tissue from normal subjects ($n = 3$ samples) and patients with DKD ($n = 3$ samples). Scale bar, 50 µm.

Representative western blot of OTUD5 expression in renal cortex of T2DM (**f**) mice and T1DM mice (**g**). ($n = 6$ samples). Representative western blot of OTUD5 expression in renal cortex of NOD (**h**) mice and db/db mice (**i**). ($n = 6$ samples). **j, k** Representative western blot of OTUD5 expression in primary podocytes of T2DM (**j**) and T1DM (**k**) mice. ($n = 6$ samples). **l** MPC5 cells transfected with Flag-OTUD5 were stimulated with HG/PA for 8 h. Real-time qPCR showed the mRNA levels of *Il6* and *Tnfa*. ($n = 3$ independent experiments; $P$ values were determined by one-way ANOVA with Bonferroni's correction and data are presented as mean ± SD). Representative western blot of Cleaved Caspase3 (**m**) and Nephrin (**n**) expression in OTUD5-overexpression podocytes stimulated by HG/PA for 24 h. ($n = 3$ independent experiments).

To date, four lysine (Lys) residues in TAK1 protein, K34, K158, K209, and K562, have been identified as potential sites of K63-linked polyubiquitination[27,28] (Fig. 4i). Thus, we explored which lysine residue ubiquitination could be removed by OTUD5. We constructed the site mutant plasmids of TAK1, including K34R, K158R, K209R, and K562R. Among them, we found that the K158R mutant abolished OTUD5-mediated deubiquitination (Fig. 4j), suggesting that lysine residue K158 of TAK1 is a specific site for OTUD5 deubiquitination. Taken together, OTUD5 decreased K63-linked polyubiquitination of TAK1 at position K158 via its active site C224.

## OTUD5 negatively regulates TAK1 activation and inflammation in podocytes

Extensive reports have shown that K63-linked ubiquitinating modification is responsible for the functional regulation of substrates, rather than the stability regulation[29]. Western blot assay confirmed that OTUD5 overexpression significantly inhibited HG/PA-induced TAK1 phosphorylation in podocytes but did not change the total TAK1 protein level (Fig. 5a, Supplementary Fig. 5a). In contrast, OTUD5 knockdown by siRNA aggravated TAK1 activation in HG/PA-challenged podocytes (Fig. 5b). Similar results were confirmed in the kidney tissues of both T2DM and T1DM mice (Fig. 5c, d, Supplementary Fig. 5b, c). It is well known that the MAPK pathway is the main downstream signal of TAK1 activation and mediates the production of inflammatory cytokines[30,31]. The MAPK signal consists of three subfamilies: c-Jun NH2-terminal kinase (JNK), extracellular signal-regulated kinase (ERK), and p38[32]. Our results showed that the activation of ERK, P38, and JNK was also inhibited by OTUD5 overexpression, but exacerbated by OTUD5 knockdown in HG/PA-treated podocytes (Fig. 5e, f, Supplementary Fig. 5d, e). Next, a small-molecule TAK1-specific inhibitor Takinib was used to block the phosphorylation of TAK1. As expected, Takinib pretreatment significantly reversed the HG/PA-induced TAK1 phosphorylation and subsequent activation of the MAPKs in OTUD5-downregulated podocytes (Fig. 5g). Takinib also inhibited the mRNA levels of *Il6* and *Tnfa* in HG/PA-challenged OTUD5-deficient podocytes (Fig. 5h). Together, OTUD5 negatively regulates TAK1-MAPK activation and inflammation in podocytes under HG/PA treatment.

Then, we tried to explore how OTUD5-mediated TAK1 deubiquitination prevents TAK1 phosphorylation. As we know, TAK1 phosphorylation requires an upstream kinase TAB2. Hirata et al. have reported that K63-linked TAK1 polyubiquitination promotes the formation of the TAK1/TAB2 complex, leading to the phosphorylation of TAK1[33]. Thus, we hypothesize that OTUD5 may hinder TAK1-TAB2 interaction since it removes the K63-linked ubiquitination of TAK1. As expected, the overexpression of OTUD5 reduced the interaction between TAK1 and TAB2, while the catalytically inactivated mutant OTUD5-C224S failed (Fig. 5i). In addition, OTUD5 inhibited the activation of the TAK1-MAPK pathway in HG/PA-treated podocytes but failed when its deubiquitinating activity was canceled in the C224S mutant (Fig. 5j). We further examined whether OTUD5-mediated TAK1 inactivation depends on TAK1 deubiquitination at the K158 site. As

shown in Fig. 5k, the interaction between TAK1 and TAB2 was reduced when K158 in TAK1 was mutated into R residue, while OTUD5 failed to further affect the interaction between TAK1 and TAB2. Similarly, the TAK1 K158R mutation inhibited TAK1-MAPK activation in response to HG/PA stimulation (Fig. 5l). These findings elucidate that OTUD5 inhibits TAK1 activation by blocking TAK1-TAB2 interaction via OTUD5-mediated TAK1 deubiquitination at the K158 site.

## Inhibition of TAK1 eliminates the aggravated podocyte injury and DKD in OTUD5CKO-T2DM mice

To confirm the OTUD5-TAK1 regulatory axis in vivo, we used a specific TAK1 inhibitor Takinib to gavage *Otud5*$^{fl/fl}$-T2DM mice and OTUD5CKO-T2DM mice (Fig. 6a). Takinib did not affect the profile of blood glucose levels in T2DM mice (Fig. 6b). Takinib significantly inhibited the aberrant activation of the TAK1-MAPK cascade in the kidneys of OTUD5CKO-T2DM mice (Fig. 6c). Biochemical analysis (Fig. 6d–f) and histochemical stainings (Fig. 6g) revealed that Takinib ameliorated both functional and structural injuries of OTUD5CKO-T2DM mouse kidneys. Electron microscopy (Fig. 6h–k) and immunofluorescence staining (Fig. 6l) showed that Takinib significantly mitigated the exacerbated podocyte injury in OTUD5CKO-T2DM mice. The mRNA levels of inflammatory cytokines showed similar changing trends (Supplementary Fig. 6a, b). These findings demonstrate that podocyte OTUD5 deficiency fails to aggravate podocyte injury when TAK1 is inhibited, indicating that TAK1 activation mediated the effect of OTUD5 deletion on DKD pathology.

## Podocyte-specific overexpression of OTUD5 alleviates podocyte injury and DKD in T2DM mice

Finally, we tested the therapeutic effect of podocyte-specific overexpression of OTUD5 against DKD in T2DM mice. To accomplish this, we constructed AAV9 encoding podocyte-specific promotor (NPHS1) and OTUD5 particles (AAV-OTUD5) in *Otud5*$^{fl/fl}$ mice and injected it through the tail vein (Fig. 7a). Western blot and immunofluorescence IF analysis confirmed successful overexpression of OTUD5 in podocytes (Supplementary Fig. 7a, b). We observed that OTUD5 overexpression had no impact on blood glucose levels in mice (Fig. 7b). Firstly, compared to the control group where mice were injected with the control particle AAV-EV, mice injected with AAV-OTUD5 showed significantly reduced levels of TAK1 ubiquitination and phosphorylation (Fig. 7c, d), which were consistent with the results observed in cells. As expected, AAV-mediated overexpression of OTUD5 in podocytes effectively attenuated renal injuries (Fig. 7e–h, Supplementary Fig. 7c, d) and podocyte injuries (Fig. 7i–m) in the kidneys of T2DM mice, evidenced by the same examinations. Collectively, podocyte-specific overexpression of OTUD5 alleviates podocyte injury and DKD in T2DM mice, accompanied by decreased TAK1 ubiquitination and phosphorylation in kidneys.

## Discussion

In this study, we observed the downregulation of OTUD5 in both HG/PA-challenged podocytes and diabetic renal tissue. Podocyte-specific

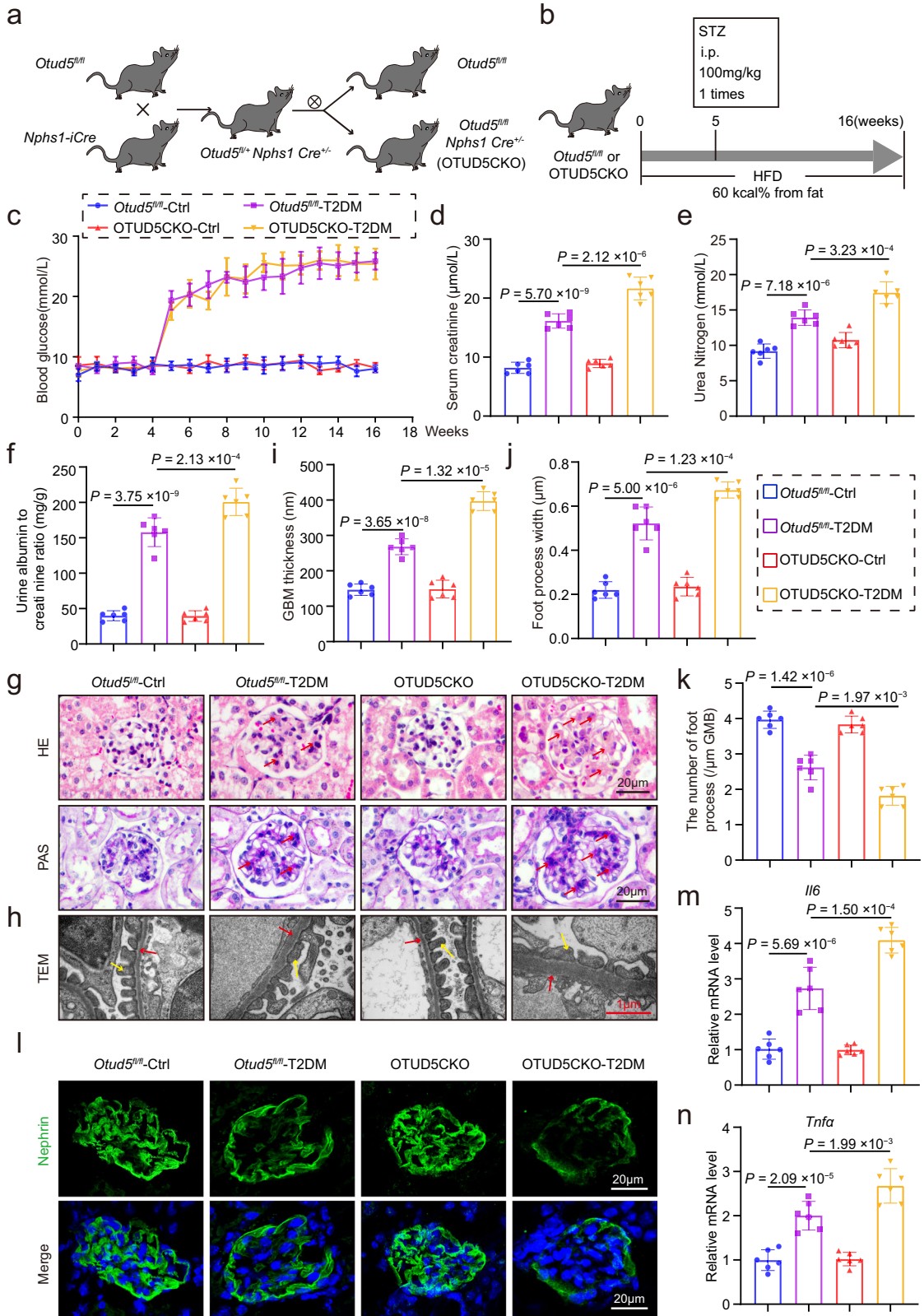

**Fig. 2 | Podocyte-specific *Otud5* knockout aggravates podocyte injury and DKD in T2DM mice. a** Schematic diagram of the strategy for the generation of podocyte-specific *Otud5* knockout mice (OTUD5CKO). **b** Schematic diagram depicting the procedure of STZ/HFD-induced T2DM mice. **c** Weekly monitoring of blood glucose levels in mice. Data are presented as mean ± SD. The levels of serum creatinine (**d**), urea nitrogen (**e**), and urine albumin to creatinine ratio (**f**) were analyzed in mice. **g, h** Representative images of hematoxylin and eosin staining (H&E), periodic acid-Schiff (PAS), and transmission electron microscopy (TEM) in mice. Scale bar: black 20 μm, red 1 μm. (*n* = 6 samples). Quantification of glomerular basement membrane (GBM) thickness (**i**) and podocyte foot process numbers (**j, k**) in the glomeruli. **l** Representative immunofluorescence (IF) images of Nephrin expression in glomeruli from mice. Scale bar, 20 μm. (*n* = 6 samples). Real-time qPCR showing mRNA levels of *Il6* (**m**) and *Tnfα* (**n**) in kidney tissues of each group. *n* = 6 for each group. For **d**–**f**, **i**–**k**, **m**, and **n**, *P* values were determined by one-way ANOVA with Bonferroni's correction, and data are presented as mean ± SD.

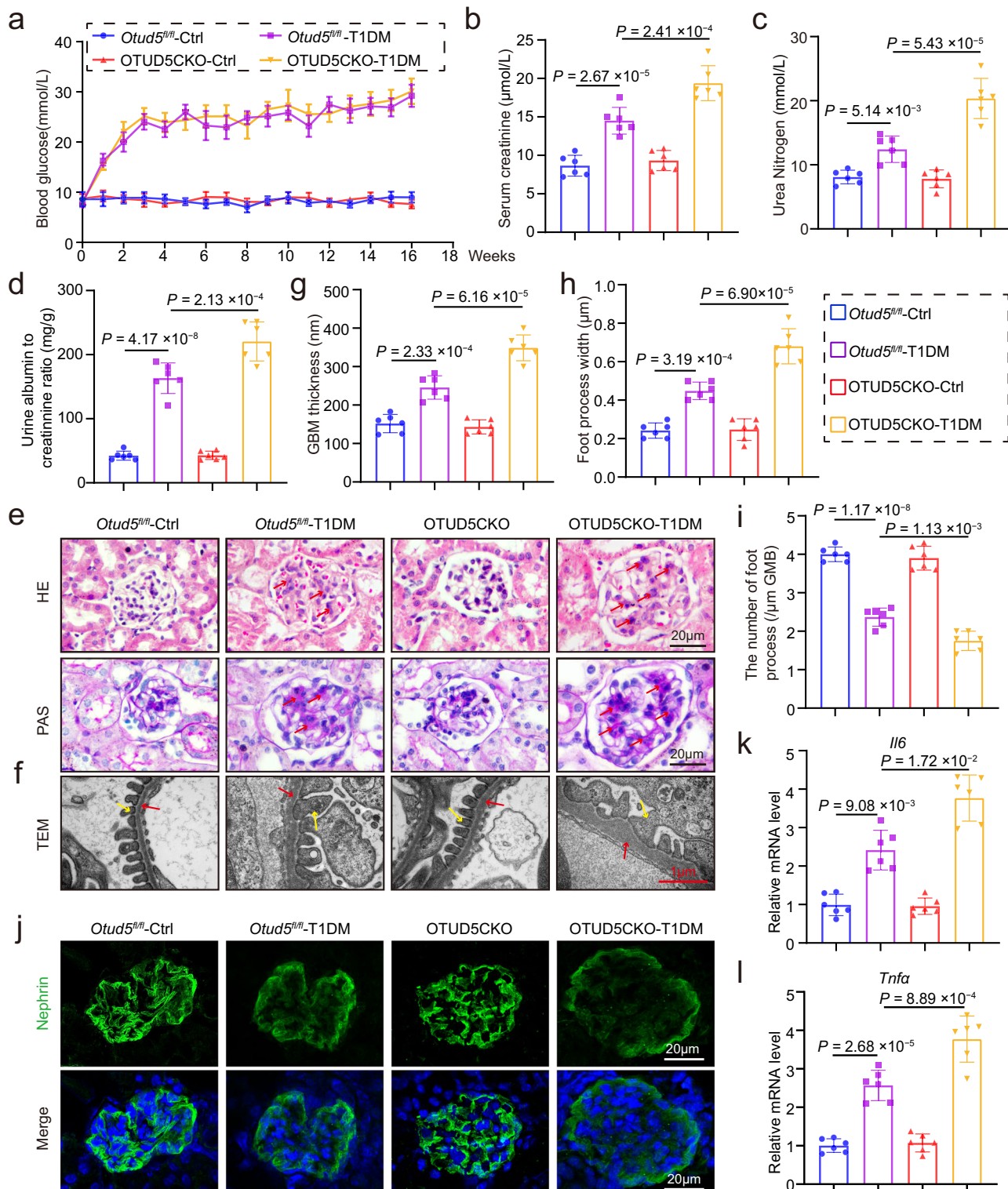

**Fig. 3 | OTUD5CKO exacerbates podocyte injury and DKD in T1DM mice.**
**a** Weekly monitoring of blood glucose levels in mice. Data are presented as mean ± SD. The levels of serum creatinine (**b**), urea nitrogen (**c**), and urine albumin to creatinine ratio (**d**) were analyzed in mice. **e, f** Representative images of H&E, PAS, and TEM in mice. Scale bar: black 20 μm, red 1 μm. (*n* = 6 samples). Quantification of GBM thickness (**g**) and podocyte foot process numbers (**h, i**) in the glomeruli. **j** Representative IF images of Nephrin expression in glomeruli from mice. Scale bar, 20 μm. (*n* = 6 samples). Real-time qPCR showing mRNA levels of *Il6* (**k**) and *Tnfα* (**l**) in the kidney tissues of each group. *n* = 6 for each group. For **b**–**d**, **g**–**i**, **k**, and **l**, *P* values were determined by one-way ANOVA with Bonferroni's correction, and data are presented as mean ± SD.

*Otud5* knockout significantly exacerbated podocyte injury and DKD in diabetic mice. Conversely, the overexpression of OTUD5 in podocytes ameliorated podocyte injury and DKD in T2DM mice. Mechanistically, we demonstrated that OTUD5 directly interacts with TAK1 and reduces

K63-linked polyubiquitination of TAK1 at residue K158 through its active site C224, subsequently inhibiting the TAK1-mediated inflammatory response and injury in podocytes. Finally, we verified the role of the OTUD5-TAK1 axis in regulating podocyte injury and DKD by

inhibiting TAK1 in vivo. Our study reports the role of OTUD5 in podocyte injury, expanding the understanding of the pathogenesis of DKD.

Podocytes are essential for maintaining the integrity and function of the glomerular filtration barrier, which is crucial for renal function[7]. Podocyte injury leads to glomerulosclerosis, proteinuria, and eventually renal failure[8]. Importantly, podocyte injury is a key factor contributing to proteinuria in DKD, and its significant role in the pathogenesis of DKD has attracted considerable attention[6]. Although podocytes are relatively few in number, their role in DKD pathogenesis is of utmost importance. Zhong et al. demonstrated that Arctigenin attenuates DKD through the activation of protein phosphatase 2A and then inhibiting NF-κB-mediated inflammatory response in podocytes[34]. Podocyte-specific deletion of claudin-5, a crucial regulator of podocyte function, exacerbates podocyte injury and DKD in diabetic mice[35]. Therefore, targeting key proteins in podocytes and protecting podocytes hold promise for slowing the progression of DKD. Podocyte inflammation is one of the key pathways contributing to podocyte injury in DKD. Injured podocytes can release inflammatory cytokines and chemokines that induce the recruitment of immune cells in the kidney. Additionally, podocytes also express receptors for various inflammatory cytokines, which promote the occurrence and development of renal inflammation[36]. Jiang et al. reported that attenuation of podocyte inflammatory response under HG stimulation protected podocytes from injuries[37]. Here, we elucidated that OTUD5 in podocytes plays a protective role against diabetic kidney injury by inhibiting TAK1-mediated inflammation and injury, further highlighting the critical role of podocyte injury in DKD.

In recent years, considerable evidence suggests that DUBs are involved in the development of DKD. It is worth noting that the role of DUBs may vary in different cell types of DKD. For instance, USP25 has been shown to ameliorate DKD by inhibiting TRAF6-mediated inflammatory responses in glomerular mesangial cells and kidney-infiltrating macrophages[38]. USP9X alleviates diabetic renal fibrosis by inhibiting epithelial-to-mesenchymal transformation in tubular epithelial cells[39] and blocking the activation of the TGF-β/Smad pathway in mesangial cells[40]. In contrast, USP22 exacerbates tubulointerstitial fibrosis progression through deubiquitinating and stabilizing Snail1 in diabetic mice[16]. Considering the significance of podocytes in DKD, we performed RNA-seq on HG/PA-challenged podocytes and identified a functional DUB, OTUD5, in podocyte inflammation and injury. Podocyte-specific *Otud5* knockout in diabetic mice aggravated podocyte injury and DKD, while overexpression of OTUD5 in podocytes protects podocytes and kidneys in diabetic mice. From a spatial expression pattern perspective, OTUD5 may exist in different cell types throughout the entire renal parenchyma. Interestingly, Chu et al. have recently reported that renal tubular epithelial cell-specific OTUD5 knockout renders kidneys susceptible to I/R injury, while AAV-mediated OTUD5 therapy protects renal function against I/R injury[41]. In this study, we focus on the crucial role of OTUD5 in podocytes. However, OTUD5 in other renal cell types may be also involved in the pathogenesis of DKD, which warrants further investigation. Anyway, our study, combined to Chu's work[41], strengthens the importance of OTUD5 in the kidney and highlights maintaining OTUD5 as a promising therapy for renal diseases.

TAK1, a member of the MAP3K family, is a key protein involved in pro-inflammatory signal transduction and plays a crucial role in podocyte injury. For instance, inhibition of TAK1 phosphorylation by activation of GPR120 in podocytes alleviated podocyte inflammation and DKD in diabetic mice[25]. It has been reported that small-molecule TAK1 inhibitors can block the occurrence and development of DKD[42]. However, TAK1 also has physiologic roles in podocytes. Podocyte-specific deletion of TAK1 in mice has been reported to reduce the expression of WT1 and Nephrin, resulting in the disruption of podocyte structure and loss of foot processes[43]. In addition,

TAK1 shows a wide distribution in humans and regulates multiple cellular actions including cell survival, growth, immune response, and metabolism. Therefore, complete deletion of TAK1 or whole-body inhibition of TAK1 activity may also induce serious side effects[44]. In the present study, Takinib that inhibited the whole-body TAK1 showed a strong basal phenotype in OTUD5[f/f]-T2DM mice, which may make it very difficult for podocyte OTUD5 deficiency to aggravate the DKD phenotypes in Takinib-treated mice. It may be a limitation of this study. The use of podocyte-specific TAK1 inhibition/knockout may be preferable to investigate the podocyte OTUD5-TAK1 axis in podocytes in DKD. Anyway, our study provides a potential approach to prevent the pathological TAK1 signaling in podocytes through the manipulation of OTUD5. Specifically, podocyte-specific overexpression of OTUD5 potentially serves as a feasible strategy to inhibit podocyte inflammation by down-regulating TAK1 activity, ultimately controlling the progression of DKD. Thus, our finding on OTUD5-TAK1 axis in podocytes shed light on a TAK1-targeting strategy for DKD treatment.

Emerging studies have demonstrated that ubiquitination is functionally involved in the regulation of TAK1 activity[45,46]. Li et al. reported that TRIM8, an E3 ubiquitin ligase, promoted the K63-linked polyubiquitination of TAK1 to mediate TNF-α/IL-1β-induced TAK1 activation[28]. The deubiquitinating enzyme CYLD interacts directly with TAK1 and removes its K63-linked polyubiquitin chain, which blocks the activation of TAK1 and downstream JNK-p38 cascade[47]. Here, we elucidated that OTUD5 inhibits TAK1 activation by removing the K63-linked ubiquitin from TAK1. Furthermore, we identified the lysine residue K158 on TAK1 as a critical site for the deubiquitination by OTUD5. It is worth noting that there is no reduction in ubiquitination level of K158R mutant compared to that of wide-type TAK1. As we know, ubiquitination/deubiquitination happens at many lysine sites in TAK1. It is possible that the ubiquitination at K158 accounts for only a small portion of the total TAK1 ubiquitination or that the ubiquitination of TAK1 at other lysine sites is increased to compensate for the loss of ubiquitination at the K158 site. It is well established that the formation of the TAK1-TABs complex is required for the activation of TAK1[33]. Here, we demonstrate that OTUD5-mediated K158 deubiquitination of TAK1 prevents TAK1-TAB2 interaction and then TAK1 phosphorylation. This suggests that the formation of the TAK1-TAB2 complex may be a dynamically regulated process, and the ubiquitin-mediated conformational change of TAK1 influences its binding to TAB2. This finding identifies OTUD5 as an inhibitor of TAK1 activation, which may serve as an alternative therapeutic strategy to regulate TAK1 hyperactivation.

A limitation of this study may be the unclear mechanism by which diabetes or HG/PA downregulates OTUD5 expression in podocytes. We made preliminary predictions of OTUD5 promoter-binding transcription factors (TFs) via the Transcription factor prediction website (http://genome.ucsc.edu/). Some TFs were predicted to regulate OTUD5 transcription, such as ZNF384, ESRRB, and ESRRA. It will be interesting to further investigate whether OTUD5 downregulation in HG/PA-challenged podocytes is related to these TFs. In addition, we acknowledge that the alterations in albuminuria, creatinine, and BUN in our diabetic model did not reach the criteria for an ideal mouse model of diabetic nephropathy[48]. In our study, we mainly focus on podocyte loss, which is an early event in DKD development. Therefore, we selected a relatively short 4-month hyperglycemia modeling to facilitate the observation of podocyte loss. This shorter duration of hyperglycemia may account for the relatively weak changes in biochemical indicators of DKD. We do think that it would be more compelling to further validate the role of OTUD5 in a DKD model with longer hyperglycemia duration and more severe kidney damage. Overall, we reveal an OTUD5-TAK1 axis in podocyte inflammation and injury and highlight the potential of OTUD5 as a promising therapeutic target for DKD.

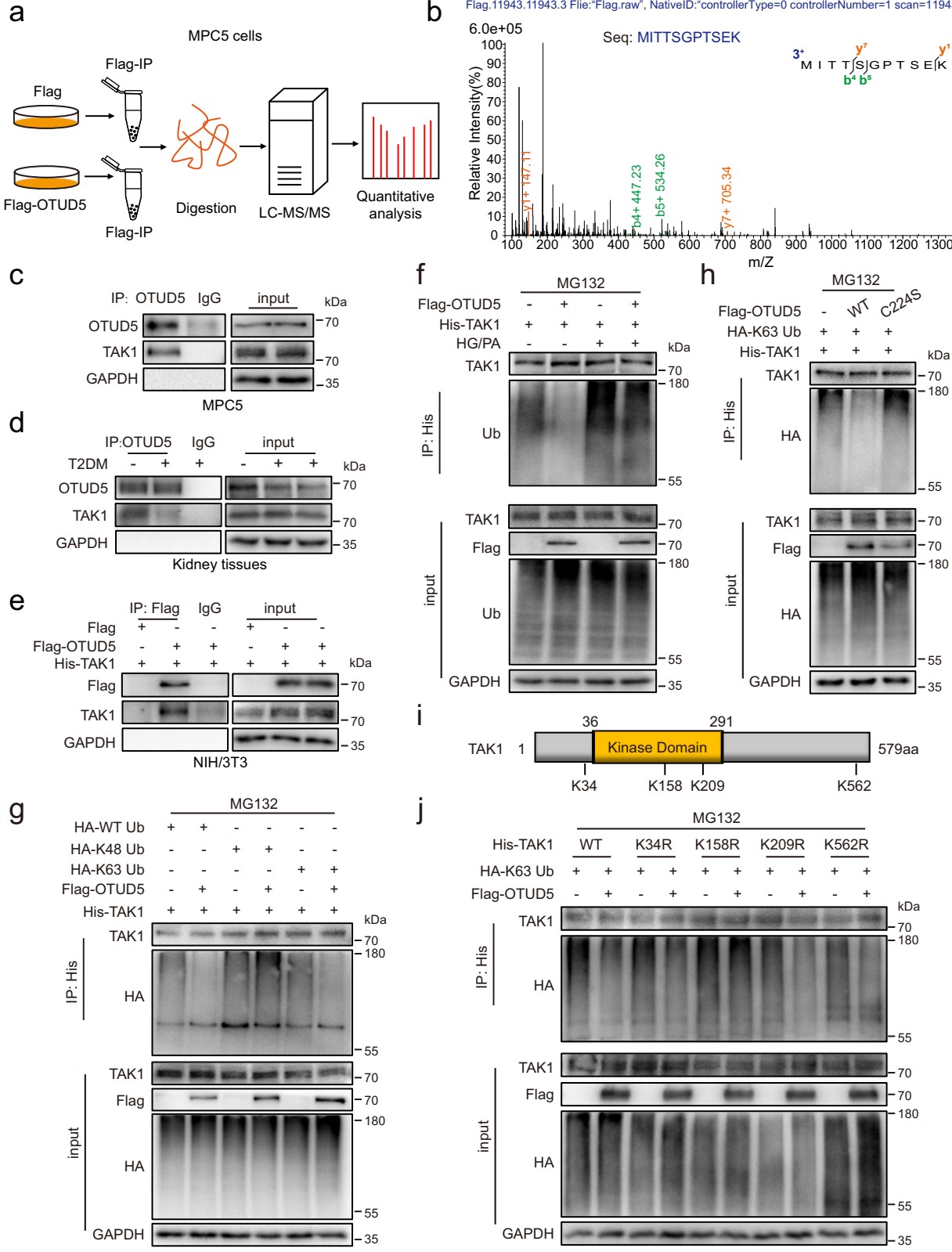

## Methods

### Antibodies

Antibodies against OTUD5 (#21002-1-AP), FLAG-Tag (#20543-1-AP), HA-Tag (#51064-2-AP), His-Tag (#66005-1-IG), Rabbit IgG (#30000-0-AP), Mouse IgG (#B900620) were purchased from Proteintech (Wuhan, China). Antibodies against ERK (#sc-514302), P-ERK (#sc-7383), JNK (#sc-7345), and P-JNK (#sc-6254) were purchased from Santa Cruz (CA, USA). Antibodies against TAK1 (#5206S), P-TAK1 (Ser412) (#9339S), TAB2 (#3745S), Ubiquitin (P4D1) (#3936), P38 (#8690S) and P-P38 (#4631S) were purchased from Cell Signaling Technology (MA, USA). Antibodies against GAPDH (#MB001) were purchased from Bioworld (Missouri, USA).

**Fig. 4 | Identification of TAK1 as a potential substrate protein of OTUD5.**
**a** Schematic illustration of a quantitative proteomic screen to identify proteins binding to OTUD5. **b** Mass spectrometry/mass spectrometry (MS/MS) spectrum of the peptide MITTSGPTSEK from TAK1. Co-immunoprecipitation (Co-IP) of OTUD5 and TAK1 in MPC5 cells (**c**) and kidney tissues (**d**). Endogenous OTUD5 was immunoprecipitated. (*n* = 3 independent experiments). **e** Co-IP of OTUD5 in NIH/3T3 co-transfected with Flag-OTUD5 and His-TAK1 plasmids. Exogenous OTUD5 was immunoprecipitated using an anti-Flag antibody. (*n* = 3 independent experiments). **f** His-TAK1 and Flag-OTUD5 were transfected into MPC5 cells with or without HG/PA treatment and then subjected to 10 μM MG132 for 6 h. Ubiquitinated TAK1 was detected by immunoblotting using an anti-ubiquitin antibody. (*n* = 3 independent experiments). **g** His-

TAK1, HA-WT Ub, HA-K48 Ub, and HA-K63 Ub were transfected into NIH/3T3 together with Flag-OTUD5 and then subjected to 10 μM MG132 for 6 h. Ubiquitinated TAK1 was detected by immunoblotting using an anti-HA antibody. (*n* = 3 independent experiments). **h** His-TAK1 and HA-K63 Ub were transfected into NIH/3T3 together with Flag-OTUD5 (WT or C224S) and then subjected to 10 μM MG132 for 6 h. Ubiquitinated TAK1 was detected by immunoblotting using an anti-HA antibody. (*n* = 3 independent experiments). **i** Schematic illustration of the construct for mutating the ubiquitinated lysine residue of TAK1. **j** His-TAK1 (WT, K34R, K158R, K209R or K562R) and HA-WT Ub were transfected into NIH/3T3 together with Flag-OTUD5 and then subjected to 10 μM MG132 for 6 h. Ubiquitinated TAK1 was detected by immunoblotting using an anti-HA antibody. (*n* = 3 independent experiments).

## Animal experiments

All animal care and handling procedures were conducted according to the National Institutes of Health (USA) guidelines and approved by the Wenzhou Medical University Animal Policy and Welfare Committee (Approval number: Wydw2021-0182). Podocyte-specific *Otud5* knock-out Mice (Nphs1-Cre OTUD5fl/fl mice; OTUD5CKO) were obtained from Gempharmatech Co., Ltd (Jiangsu, China). The genotype of OTUD5CKO was maintained by crossing the C57BL/6JGpt-Otud5em1Cflox/Gpt mouse (OTUD5fl/fl, strain No. T052115) and the C57BL/6JGpt-H11em1Cin(Nphs1-iCre)/Gpt mouse (Myh6-iCre, strain No. T005680). The mice were housed in an environmentally controlled room at 22 ± 2.0 °C and 50% ± 5% humidity, with a 12-hour light/dark cycle, and were fed standard rodent chow and tap water. All animal experiments were conducted in a blinded manner.

(1) Streptozotocin (STZ)-induced type 1 diabetic kidney disease in mice. Six-week-old male OTUD5CKO mice and their littermate OTUD5fl/fl mice were randomly divided into 4 groups: OTUD5fl/fl + sham group, OTUD5fl/fl + STZ group, OTUD5CKO + sham group, OTUD5CKO + STZ group (*n* = 6). STZ was purchased from Sigma-Aldrich (S0130, MO, USA). Low-dose STZ (50 mg/kg) dissolved in 0.1 M sodium citrate buffer was intraperitoneally injected for 5 continuous days, while an equal amount of sodium citrate buffer (pH 4.5) was used as the control. Blood glucose (BG) levels were measured and recorded using a glucometer (One Touch Ultra Easy, Life Scan, USA), with fasting blood glucose (FBG) levels above 16.7 mmol/L considered indicative of diabetes. All mice were given free access to food and water for 16 weeks.

(2) High-fat diet feeding plus STZ injection (HFD/STZ) induced type 2 diabetic kidney disease in mice. Six-week-old male OTUD5CKO mice and their littermate OTUD5fl/fl mice were randomly divided into 4 groups: OTUD5fl/fl + sham group, OTUD5fl/fl + HFD/STZ group, OTUD5CKO + sham group, and OTUD5CKO + HFD/STZ group (*n* = 6). The high-fat diet (HFD, 60 kcal% from fat) was purchased from Research Diets (D12492, USA). A single dose of STZ (100 mg/kg) was injected after 4 weeks of feeding the HFD or a control diet. The mice continued to be maintained on an HFD or control diet for another 12 weeks.

(3) Pharmacological inhibition of TAK1 in type 2 diabetic mice. Six-week-old male OTUD5CKO mice and their littermate OTUD5fl/fl mice were randomly divided into 4 groups: OTUD5fl/fl + T2DM group, OTUD5fl/fl + T2DM + Takinib group, OTUD5CKO + T2DM group, and OTUD5CKO + T2DM + Takinib group (*n* = 6). The TAK1 inhibitor, Takinib, was purchased from MedChemExpress (HY-103490, California, USA). Takinib (50 mg/kg) was dissolved in Sodium carboxymethyl cellulose (CMC-Na) and administered orally every other day.

(4) Podocyte-specific over-expression of OTUD5 in type 2 diabetic mice. The recombinant adeno-associated virus serotype 9 (AAV9) carrying OTUD5 or EV cDNA with a podocyte-specific promoter NPHS1 (Bioqure Genetech, Hangzhou, China) was used for over-expression of OTUD5 or control. Six-week-old male OTUD5fl/fl mice were randomly divided into 2 groups: OTUD5fl/fl + T2DM + AAV-EV group and OTUD5fl/fl + T2DM + AAV-OTUD5 group (*n* = 6). AAV9 was administered via the tail vein at a dose of 2E + 11 vg.

Before sacrifice, mice were euthanized under anesthesia using pentobarbital sodium (50 mg/kg, intraperitoneally). Blood samples and 24-hour urine samples were collected for biochemical analysis. Kidney tissues were harvested for histopathological analysis. Protein or RNA was extracted from kidney tissue for subsequent analysis.

## Cell culture and transfection

The mice renal podocyte cell line (MPC5) was obtained from the Cell Bank of the Chinese Academy of Sciences (Shanghai, China). MPC5 cells were cultured in RPMI 1640, 10% fetal bovine serum (Meilunbio, PWL001) and 10 IU/mL of recombinant murine c-interferon (IFN-γ, Invitrogen, CA, USA) at 33 °C and 5% CO$_2$. To induce differentiation, the cells were transferred to 37 °C for 10–14 days and the medium was replaced without IFN-γ. The NIH/3T3 cell line was purchased from the Shanghai Institute of Biochemistry and Cell Biology (Shanghai, China) and cultured in DMEM containing 4.5 g/L glucose. The culture medium was supplemented with 10% fetal bovine serum (Meilunbio, PWL001), 100 U/mL penicillin, and streptomycin (Sbjbio, Nan Jing, BC-CE-007) at 37 °C with 5% CO$_2$.

Primary podocytes were extracted as described below. The kidney cortex was finely chopped into 1 mm3 pieces and then digested in collagenase IV buffer (C8160, Solarbio, Beijing, China) at 37 °C for 15 minutes. The digested tissue was subsequently passed through cell filters of decreasing sizes: 150 μm, 75 μm, and finally 40 μm. After a precipitation period of 1–2 minutes, large debris tubules adhered to the bottom of the petri dish, while most of the glomeruli floated and small debris tubules were present in the supernatant. The supernatant was then passed through the 40 μm filter again, repeating this step until a highly purified glomerular supernatant was obtained. After 7 days, the glomerular cells were digested and then filtered with a 40 μm cell filter. Extracted primary podocytes were further incubated.

Gene expression was silenced using small interfering RNA (siRNA) purchased from GenePharma (Shanghai, China). MPC5 cells were transfected with siRNA targeting OTUD5 (sense: GGACAUGCAU-GAGGUUGUUTT, anti-sense: AACAACCUCAUGCAUGUCCTT) with LipofectAMINE™ 2000 (#L3000015, Thermo Fisher Scientific, German).

The overexpression plasmids (Flag-OTUD5-WT, Flag-OTUD5-C224A, His-TAK1-WT, His-TAK1-K34R, His-TAK1-K158R, His-TAK1-K209R, and His-TAK1-K562R) were obtained from Tsingke Bio-technology Co., Ltd. (Beijing, China). Plasmids (HA-Ub, HA-K48 Ub, and HA-K63 Ub) were obtained from Addgene. The above plasmids were transfected by LipofectAMINE™3000 (L3000150, Thermo Fisher Scientific, German).

## Biochemical analysis of serum and urine samples

Serum creatinine, urea nitrogen, and urine creatinine levels were measured by enzymatic assays using an auto-chemistry analyzer. Urine albumin was detected using an ELISA kit (H127-1-1, Nanjing Jiancheng Bioengineering Institute, Nanjing, China) according to the manufacturer's instructions.

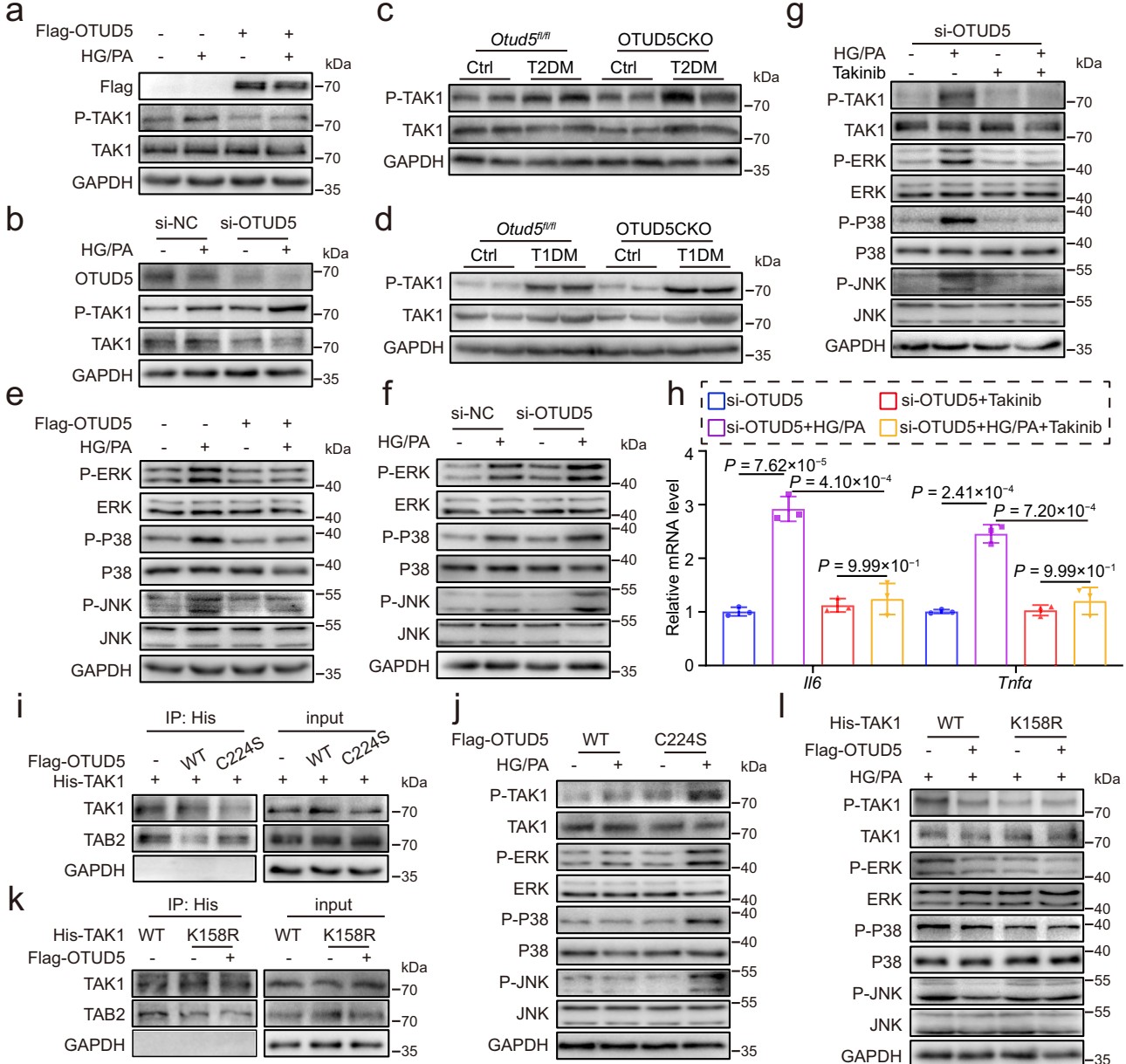

**Fig. 5 | OTUD5 negatively regulates TAK1 activation and inflammation in podocytes.** MPC5 cells transfected with Flag-OTUD5 (**a**) or si-OTUD5 (**b**) were stimulated with HG/PA for 30 min. Representative western blot analysis of P-TAK1. (*n* = 3 independent experiments). **c, d** Representative western blot analysis of P-TAK1 in kidney tissues of each group. (*n* = 6 samples). MPC5 cells transfected with Flag-OTUD5 (**e**) or si-OTUD5 (**f**) were stimulated with HG/PA for 30 min. Representative western blot analysis of phosphorylated and total protein levels of ERK, P38, and JNK. (*n* = 3 independent experiments). MPC5 cells transfected with si-OTUD5 were pretreated with 10 μM Takinib (TAK1 inhibitor) for 1 h before exposure to HG/PA. **g** Levels of P-TAK1, P-ERK, P-P38, and P-JNK were detected by western blot. **h** Realtime qPCR showing mRNA levels of *Il6* and *Tnfα*. (*n* = 3 independent experiments; *P* values were determined by one-way ANOVA with Bonferroni's correction and data are presented as mean ± SD). **i** His-TAK1 was transfected into NIH/3T3 with or without Flag-OTUD5 (WT or C224A). Co-IP was performed with an anti-His antibody, followed by a western blot of TAK1 and TAB2. (*n* = 3 independent experiments). **j** MPC5 cells transfected with Flag-OTUD5 (WT or C224A) were stimulated with HG/PA for 30 min. Representative western blot analysis of phosphorylated and total protein levels of TAK1, ERK, P38, and JNK. (*n* = 3 independent experiments). **k** His-TAK1(WT or K158R) was transfected into NIH/3T3 with or without Flag-OTUD5. Co-IP was performed with an anti-His antibody, followed by a western blot of TAK1 and TAB2. (*n* = 3 independent experiments). **l** His-TAK1 (WT or K158R) and Flag-OTUD5 were transfected into MPC5 cells for 24 h and then stimulated by HG/PA for 30 min. Representative western blot analysis of phosphorylated and total protein levels of TAK1, ERK, P38, and JNK. (*n* = 3 independent experiments).

## Histological analysis
Mice kidney tissues were fixed with 4% paraformaldehyde and embedded in paraffin. Hematoxylin and eosin (H&E, G1120, Solarbio, Beijing, China) staining of tissue sections (5 μm) were used to evaluate the histopathological damage. Periodic acid-Schiff (PAS, G1285, Solarbio, Beijing, China) staining was used to evaluate the mesangial dilation in the glomeruli. Images were taken with a light microscope (Nikon, Tokyo, Japan).

## Transmission Electron Microscopy
Electron microscope samples were treated and observed by the electron microscope Laboratory of Wenzhou Medical University. The renal cortex was cut into pieces (<1 mm²) and fixed with 2.5% glutaraldehyde at 4 °C. The tissue slices were washed in 0.2 mol/L PBS solution and then incubated with 1% osmic acid at room temperature for 2 hours. After a gradient dehydration process using ethanol, the sections were embedded in Pon812 resin with acetone

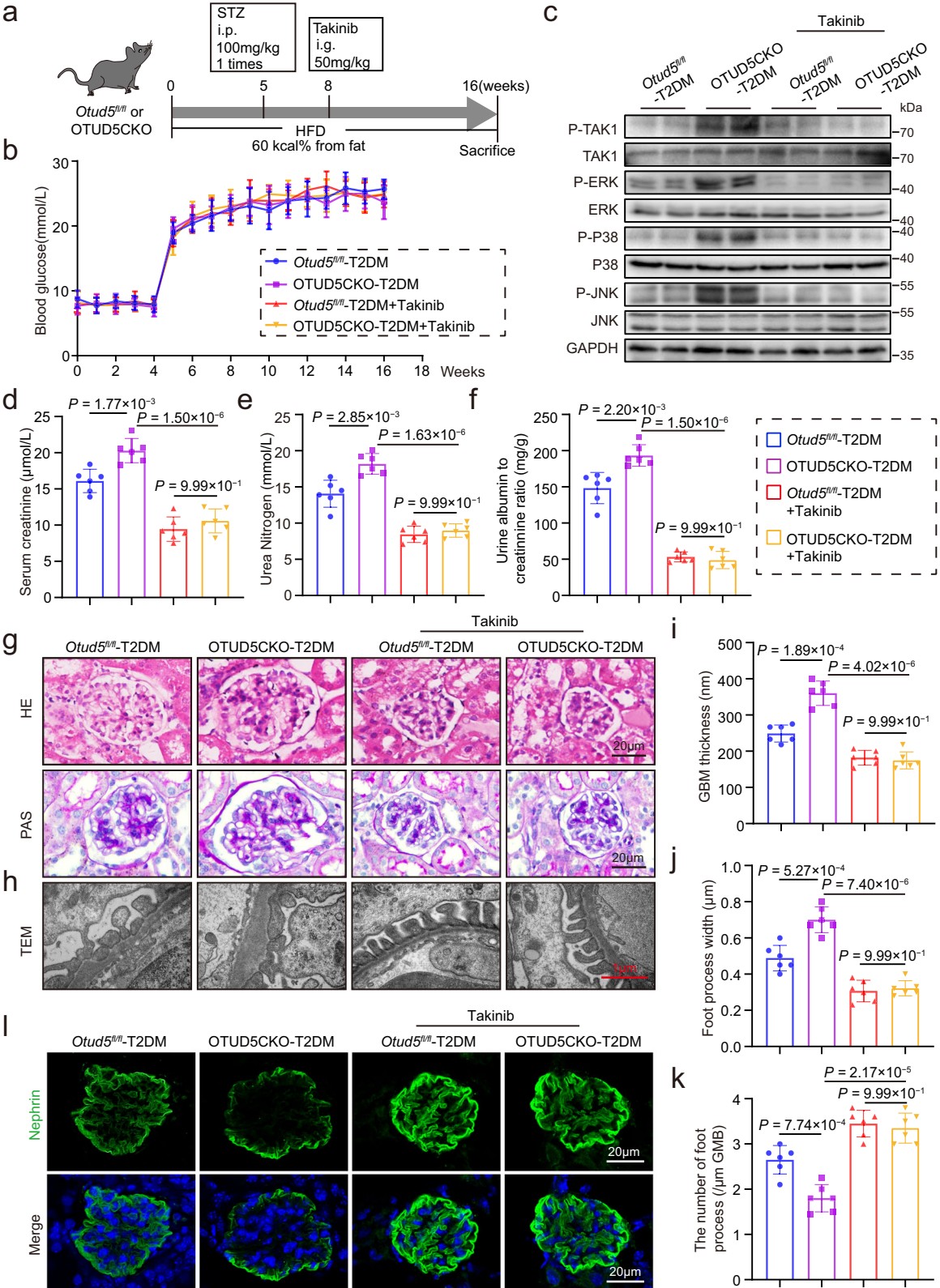

**Fig. 6 | Inhibition of TAK1 eliminates the aggravated podocyte injury and DKD in OTUD5CKO-T2DM mice. a** A schematic diagram illustrating the procedure of T2DM-induced OTUD5CKO and *Otud5*^fl/fl^ mice, with or without Takinib administration. **b** Weekly measurements of blood glucose levels of the mice in the indicated groups. Data are presented as mean ± SD. **c** Representative western blot images showing phosphorylated and total protein levels of TAK1, ERK, P38, and JNK in the kidneys of indicated groups. (*n* = 6 samples). Serum creatinine (**d**), urea nitrogen (**e**), and urine albumin to creatinine ratio (**f**) levels of the mice in the indicated groups. **g** Representative H&E and PAS staining images of kidney sections. Scale bar: 20 μm. (*n* = 6 samples). **h**–**k** Representative TEM images of kidney sections and corresponding quantitative analysis. Scale bar: 1 μm. **l** Representative IF images of Nephrin expression in the glomeruli of the indicated groups. Scale bar, 20 μm. *n* = 6 for each group. For **d**–**f**, **i**, **j**, and **k**, *P* values were determined by one-way ANOVA with Bonferroni's correction, and data are presented as mean ± SD.

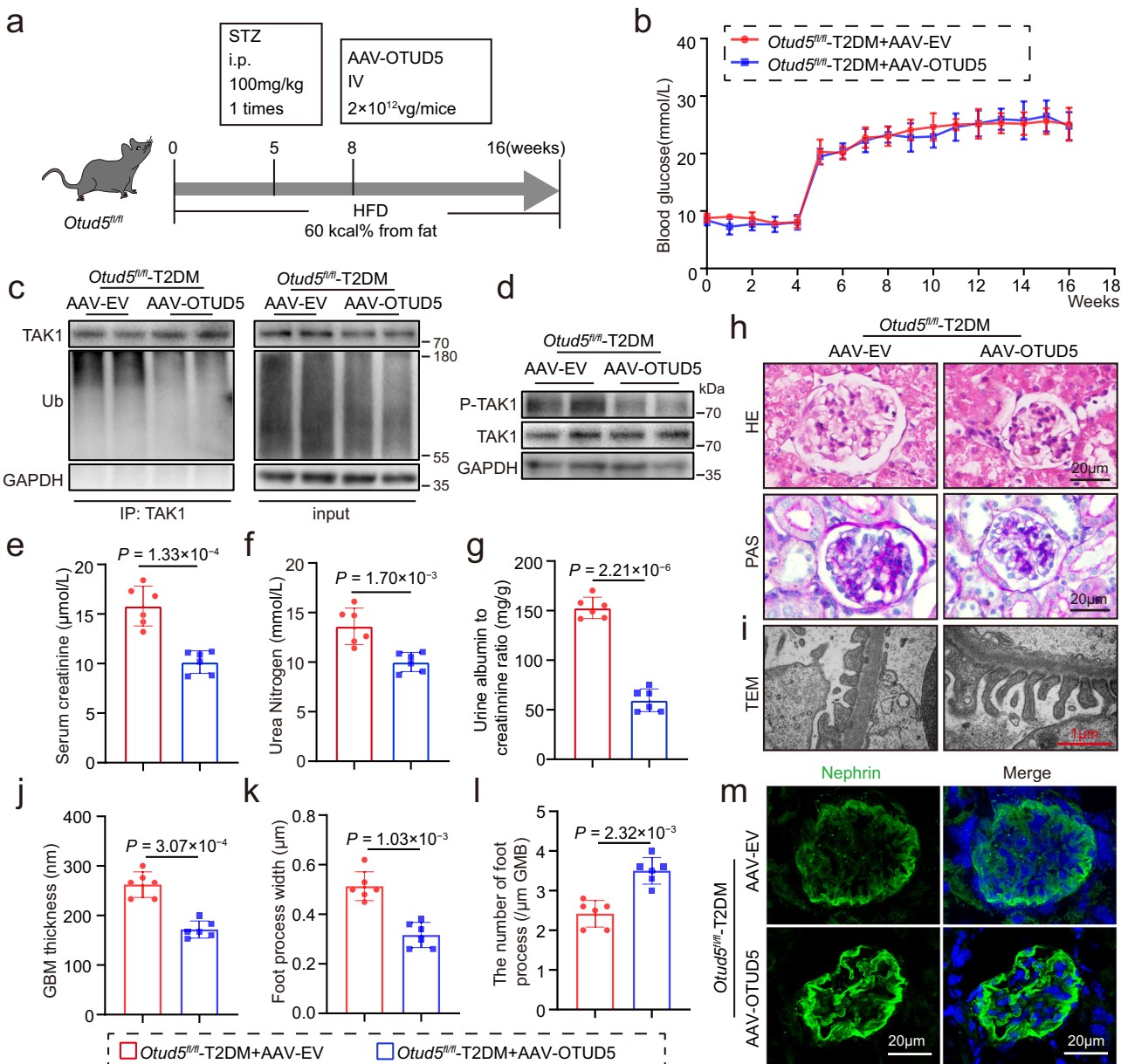

**Fig. 7 | Podocyte-specific overexpression of OTUD5 alleviates podocyte injury and DKD in T2DM mice. a** A schematic diagram illustrating the procedure of T2DM-induced *Otud5^fl/fl* mice injected with AAV-EV or AAV-OTUD5. **b** The levels of blood glucose. Data are presented as mean ± SD. **c** Ubiquitinated TAK1 was detected by immunoblotting using anti-ubiquitin antibodies in the kidneys of indicated groups. (*n* = 6 samples). **d** Representative western blot images showing phosphorylated and total protein levels of TAK1 in the kidneys of indicated groups.

(*n* = 6 samples). The levels of serum creatinine (**e**), urea nitrogen (**f**), and urine albumin to creatinine ratio (**g**). **h** Representative H&E and PAS staining of kidney sections. (*n* =6 samples). **i–l** Representative TEM images of kidney sections and corresponding quantitative analysis. Scale bar: 1 μm. **m** Representative IF images of Nephrin expression. Scale bar, 20 μm. (*n* = 6 samples). *n* = 6 for each group. For **e–g** and **j–l**, *P* values were determined by a two-tailed unpaired *t*-test, and data are presented as mean ± SD.

as solvent at 37 °C overnight. Ultrathin sections were stained with a solution of 2% uranyl acetate and lead citrate and examined under a Philips CM 120 electron microscope (Philips Medical Systems, Inc.). The thickness of the GBM and the number of foot processes were analyzed using NIH ImageJ (National Institutes of Health, Bethesda, MD).

### Transcriptome sequencing
Total RNA from untreated or HG/PA-treated MPC5 was collected using RNAiso Plus and subjected to genome-wide transcriptomic analysis by LC-Bio (Hangzhou, China). The differentially expressed genes (DEGs) were selected with a fold change >1.5 or fold change <0.667. GSEA (https://www.gsea-msigdb.org/gsea/ index.jsp) of the

signaling pathways was performed as described by LC-Bio (https://www.lc-bio.cn/).

### Western blot
Cultured cells and renal tissues were lysed in RIPA buffer (P0013C, Beyotime Biological, Shanghai, China) containing proteinase inhibitor PMSF (ST506, Beyotime Biological, Shanghai, China). Equal amounts of protein were then separated by SDS-PAGE and transferred to PVDF membranes. The membranes were subsequently blocked in 5% Bovine Serum Albumin (BSA, A8020, Solarbio, Beijing, China) and incubated with primary antibodies overnight at 4 °C. After incubating with a secondary antibody (Beyotime Biological, Shanghai, China) for 1 hour, the

membranes were detected by an enhanced chemiluminescence (ECL) system. The quantification of images was analyzed by ImageJ software.

## Co-immunoprecipitation (co-IP)
The cell and renal tissue lysates were incubated overnight with specific primary antibodies or control IgG. Protein A + G Agarose (P2012, Beyotime Biological, Shanghai, China) was added to the lysates and incubated at 4 °C for 2 hours to capture the immune complex. The supernatant was then discarded after centrifugation, and the precipitate was washed 5 times with PBS. The protein was eluted with an SDS loading buffer for western blotting.

## LC-MS/MS analysis
MPC5 cells were transfected with Flag-vector or Flag-OTUD5 plasmids. Anti-Flag and protein A + G Agarose beads were added to the cell samples. LC-MS/MS analyses were performed on a Q Exactive HF mass spectrometer that was coupled to Easy nLC (Thermo Scientific). Peptide was first loaded onto a trap column (100 μm*20 mm, 5 μm, C18) with 0.1% formic acid, then separated by an analytical column (75 μm*100 mm, 3 μm, C18)) with a binary gradient of buffer A (0.1% Formic acid) and buffer B (84% acetonitrile and 0.1% Formic acid) at a flow rate of 300 nL/min over 60 min. MS data was acquired using a data-dependent top20 method dynamically choosing the most abundant precursor ions from the survey scan (350–1800 m/z) for HCD fragmentation. The MS data were analyzed using MaxQuant software version 1.5.8.3. MS data were searched against the UniProtKB Human database (157,600 total entries, downloaded 07/2017). The trypsin was selected as a digestion enzyme. The maximal two missed cleavage sites and the mass tolerance of 4.5 ppm for precursor ions and 20 ppm for fragment ions were defined for database search. Carbamidomethylation of cysteines was defined as fixed modification, while acetylation of protein N-terminal and Lysine, and oxidation of Methionine, were set as variable modifications for database searching. The database search results were filtered and exported with <1% false discovery rate (FDR) at peptide level and protein level, respectively.

## RNA extraction and real-time quantitative PCR
Total RNA was isolated from cells or renal tissues using TRIzol (#9109, Takara, Japan), and 1ug total RNA was reverse transcribed by HiScript®III All-in-one RT SuperMix reagent (R333-01, Vazyme, Nanjing, China). The quantitative PCR was performed with SYBR Green Master Mix (Q711-02, Vazyme, Nanjing, China) in a Real-Time PCR System (QuantStudioTM 3, Thermo Fisher Scientific, CA, USA). Gene levels were normalized to β-actin, and the relative gene expression was analyzed using the $\Delta\Delta^{-Ct}$ algorithm. The primers used are listed in Supplementary Table 1.

## Human renal biopsy samples
Renal biopsies from 3 diabetic patients and 3 nondiabetic patients were collected from the Department of Pathology, the First Affiliated Hospital of Wenzhou Medical University. The non-diabetic samples were obtained from the healthy kidney poles of individuals who underwent tumor nephrectomies without diabetes or renal disease. All experiments involving human samples were approved by the Ethics Committee in Clinical Research of the First Affiliated Hospital of Wenzhou Medical University (Wenzhou, China; Approval number: 2023-0115), and informed consent was obtained from the patients. All the human study participants agreed to participate for free. All aspects of the study followed the Declaration of Helsinki of 1975, revised in 2008. The clinical characteristics of the patients are shown in Supplementary Table 2.

## Data and statistical analysis
All data were analyzed using GraphPad Prism 8.0 software (San Diego, CA, USA). Data are expressed as mean ± SD. The unpaired *t*-test was used for comparisons between two groups and one-way ANOVA analysis (Bonferroni's correction) was used for comparing multiple groups. *P* values less than 0.05 were considered statistically significant.

## Reporting summary
Further information on research design is available in the Nature Portfolio Reporting Summary linked to this article.

## Data availability
All data are included within the article or Supplementary Information or available from the authors on request. The source data are provided as a Source data file. The transcriptomic data sets generated in this study have been deposited in the Gene Expression Omnibus (GEO) database under accession code GSE254059. Source data are provided online with this paper. Source data are provided in this paper.

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

## Acknowledgements

National Natural Science Foundation of China (81930108 to G.L. and 82370829 to H.Z.) and Zhejiang Provincial Key Scientific Project (2021C03041 to G.L.).

## Author contributions

Y.Z., S.F., and G.L. drafted and revised the manuscript. Y.Z., S.F., H.Z., Q.Z., Z.F., D.X., W.L. and G.L. designed and performed the experiments. H.Z., L.L., X.H., G.W., and J.M. generated podocyte-specific *Otud5* knockout mice. Y.Z., S.F., Q.Z., Z.F., D.X., W.L., J.M., and G.L. supervised the work. All authors approved the final version of the manuscript.

## Competing interests

The authors declare no competing interests.
