## [Peer Review File · Nature Communications]

Podocyte OTUD5 alleviates diabetic kidney disease through deubiquitinating TAK1 and reducing podocyte inflammation and injuryREVIEWER COMMENTS

Reviewer #1 (Remarks to the Author):

This study explored the role of OTUD5 in diabetic Nephropathy (DN) and found that OTUD5 acts to protect DN by inhibiting podocyte inflammatory injury. Consequently, the authors identified OTUD5 interacts and stabilizes TAK1 to inhibit inflammation, leading to podocyte survival. This study addresses an interesting and important topic in the field of DN, and the study design is reasonable, and the research content is comprehensive. However, several points require clarification and additional experiments to support the authors' conclusions:

1. Line 69-73, the author described some results they did in this study, please consider if it is appropriate to display there. In addition, they could add some description of the association between post-translational modification (ubiquitination/deubiquitination) and kidney diseases.
2. Please provide detailed information on mice anesthesia
3. Line 242, we did not see any one-way ANOVA analysis in the study. Please confirm and correct.
4. In Figure 1, the addition of the experiment such as overexpression of OTUD5 in OTUD5-depleted cells to verify its on-target effects is necessary.
5. In Figure 2, the author should clarify the change of body weight of mice in different groups, as such, the muscle mass will influence the level of serum creatinine
6. In Figure 2g-h, no remarkable changes in the representative images are displayed between groups, please explain, and the addition of an arrow or enlarging the specific area will be more clear.
7. When addressing the level of BUN and creatinine, the specific time point should be mentioned, as these are the biomarkers of the injured kidney at an early stage. And more than one time-point value will be more reasonable
8. To strengthen the contribution of OTUD5 to stabilize TAK1's expression, the possibility of other degradation pathways other than ubiquitination/deubiquitination potentially mediating TAK1's expression should be excluded.
9. In Figure 7, the addition of the change in inflammatory cytokine expression will be more convincing.

Reviewer #2 (Remarks to the Author):

In their manuscript the authors investigate the role of DUBs in kidney disease. They used RNAseq to identify dysregulated DUBs and then focus on OTUD5. They use two different mouse models (STZ and STZ+HFD) to determine the role of OTUD5 in combination with an inhibitor for TAK1 (a downstream target of OTUD5) and an overexpression approach of OTUD5. To gain mechanistic insights, mutants of OTUD5 and TAK1 based on known relevant sites are introduced in vitro and the interaction of OTUD5 and TAK1 is determined. Overall, the study is well done and provides new insights. The authors need to improve some aspects in the paper:

Comments:

- The authors use the term “diabetic nephropathy”. This should be replaced by “diabetic kidney disease”.
- Abstract: The authors write that RNA-seq indicated a significantly increased expression of OTUD5. But in the image shown (Fig. 1a) it is downregulated, which is also congruent with the rest of the text.
- In the introduction the authors state that “developing podocyte specific therapies” remains a major challenge. Are any of their molecular targets podocyte specific? Is this issue addressed by their work?
- The phenotype in podocyte specific ko mice is minor. There is only a minor increase of serum creatinine, serum BUN and urine ACR.
- The observed changes in regard to the inflammatory cytokines IL6 and Tnfa are more pronounced. Does this translate into a different composition of immune cells in the kidney? Are cytokines in the urine affected?
- Fig. 6: Takinib has a strong basal phenotype. This makes it difficult to draw a conclusion.
- More methodological information on how MPC5 cells were cultured and differentiated are needed. In particular how were cells stimulated with glucose and PA? concentration, duration, change of medium etc. Details are missing.
- Were mice on a C57BL/6 “j” or “n” substrain used. Please provide this information.
- In the HFD/STZ model: please state clearly in the methods how often STZ was injected. What was the bodyweight in these mice?
- Statistics: Was correction for multiple testing performed? Were data normally distributed?

How as this ascertained?

- Fig. S2A and corresponding text: The authors state that “deletion off OTUB5 in podocytes was validated by tail genotyping”. That is not possible. There is something obviously wrong.
- In Fig. 1E the authors show a reduction of OTUD5 in the T2DM model, which is not apparent Fig. 4D.
- Can the authors explain why in Fig. 4J ubiquitination remains high in the K158R mutant? If this site is required for de-ubiquitination, as stated by the authors, I would expect less ubiquitination at baseline. Instead, there is no reduction in ubiquitination. How do the authors explain this observation?
- Fig. 5A: based on one immunoblot not statement regarding “significance” is possible. Please show multiple blots and quantification (e.g. in the supplements). Quantification is also needed for Fig. 5E,F.
- Fig. 6: Takinib has a strong basal phenotype (compare OTUD5f/f-T2DM versus OTUD5f/f-T2DM+Takinib. This may be a dominant effect, which makes it difficult to draw a conclusion. Please comment on this.
- Fig. S7A: please show markers for podocytes and other cells and also analyze other cells to demonstrate that indeed podocyte specific overexpression was achieved. This is important as the authors emphasize this aspect, e.g. in the discussion.
- A major limitation is the lack of human data. Also, no mouse model independent of STZ was used to validate their observations. At least basal phenotypes can be shown in humans and alternative mouse models.

Minor:

- The term “significant” should only be used in the context of statistical analyses; e.g. line 69 – replace “significance of podocyte injury”.

Responses to the Reviewers' comments

Reviewer #1 (Remarks to the Author):

This study explored the role of OTUD5 in diabetic Nephropathy (DN) and found that OTUD5 acts to protect DN by inhibiting podocyte inflammatory injury. Consequently, the authors identified OTUD5 interacts and stabilizes TAK1 to inhibit inflammation, leading to podocyte survival. This study addresses an interesting and important topic in the field of DN, and the study design is reasonable, and the research content is comprehensive. However, several points require clarification and additional experiments to support the authors' conclusions:

Response: Thanks for your positive comments.

Reviewer: #1-1. Line 69-73, the author described some results they did in this study, please consider if it is appropriate to display there. In addition, they could add some description of the association between post-translational modification (ubiquitination/ deubiquitination) and kidney diseases.

Response: Thank you for your valuable advice. Regarding to your suggestion on the description in lines 69-73, we think that this sentence is necessary to bring OTUD5 out. We revised this sentence: "Considering the importance of podocyte injury in DN, we performed an RNA sequencing (RNA-seq) analysis using podocytes under the high-concentration glucose and palmitic acid (HG/PA) condition and found a potentially DN-related DUB in podocytes, ovarian tumor deubiquitinase 5 (OTUD5)."

In addition, we have added more description on the association between post-translational modifications (ubiquitination/deubiquitination) and kidney diseases in the Introduction section. "For instance, the deletion of USP11 in renal tubular epithelial cells improves renal function in mouse models with renal fibrosis by promoting the degradation of EGFR (*Kidney International*, PMID: 36581018). OTUD1 and USP22 have also been shown to promote fibrosis and injury in renal tubular epithelial cells (*Acta Pharmacol Sin*, PMID:38110583; *Eur J Pharmacol*, PMID:37001578). In addition, USP13 promotes tumorigenesis of clear cell renal cell carcinoma through deubiquitinating ZHX2 (*Proc Natl Acad Sci U S A*, PMID:36037364)."

Reviewer: #1-2. Please provide detailed information on mice anesthesia

Response: Thank you. We described the detailed information on mice anesthesia in the revised Method section.

Reviewer: #1-3. Line 242, we did not see any one-way ANOVA analysis in the study. Please confirm and correct.

Response: Thank you. One-way ANOVA analysis was used in Figure 2, Figure 3, and Figure 6 to address the multiple group comparisons. In order to make it clear, we noted the respective statistical analysis method in all figure legends in the revised manuscript.

Reviewer: #1-4. In Figure 1, the addition of the experiment such as overexpression of OTUD5 in OTUD5-depleted cells to verify its on-target effects is necessary.

Response: Thank you for your suggestion. Accordingly, we overexpressed OTUD5 in OTUD5

knockdown podocytes and examined the injury and inflammation. The results shown in the new Supplementary Figure S1g-j further confirmed that overexpression of OTUD5 attenuated HG/PA-induced podocyte injury.

new Supplementary Figure S1g-j

Reviewer: #1-5. In Figure 2, the author should clarify the change of body weight of mice in different groups, as such, the muscle mass will influence the level of serum creatinine

Response: Thank you. We have shown the body weight of mice in different groups in the new Supplementary Figure 2c. As we can see, there was no significant difference in mouse body weight among four groups.

new Supplementary Figure S2c

Reviewer: #1-6. In Figure 2g-h, no remarkable changes in the representative images are displayed between groups, please explain, and the addition of an arrow or enlarging the specific area will be more clear.

Response: Thank you. We added arrows to indicate the structural changes in Figure 2g-h. In figure 2g, the degree of mesangial matrix expansion was increased in OTUD5CKO T2DM mice compared to Otud5^{fl/fl} mice. In addition, we could observe that OTUD5CKO mice exhibited larger glomerular volume in T2DM mice. In figure 2h, we noticed that the OTUD5CKO exacerbates T2DM induced glomerular basement membrane thickening (red arrow) and foot processes broadening and effacement (yellow arrow). To further quantify these changes, we also included the quantitative analysis in Figure i-k.

Reviewer: #1-7. When addressing the level of BUN and creatinine, the specific time point should be mentioned, as these are the biomarkers of the injured kidney at an early stage. And more than one time-point value will be more reasonable

Response: Thanks. In this study, blood samples were collected to measure BUN and creatinine levels at the animal experimental end when the mice were euthanized after 16 weeks of modeling. We are sorry for that we have only one time-point values for these two biomarkers.

In general, dynamic changes in these indicators are monitored clinically, rather than a single time-

point. We acknowledge that multiple measurements and comparison of values at different time points are advantageous to accurately assess kidney dysfunction.

However, in a majority of pre-clinical studies, scientists measure the mouse creatinine and BUN levels only at the end of the animal study (*Signal Transduct Target Ther*, PMID: 36450712; *Kidney Int*, PMID: 32739204; *J Am Soc Nephrol*, PMID: 36198430), which is also valuable to reflect the renal function in mice at the indicated time point.

Reviewer: #1-8. To strengthen the contribution of OTUD5 to stabilize TAK1's expression, the possibility of other degradation pathways other than ubiquitination/deubiquitination potentially mediating TAK1's expression should be excluded.

Response: Thank you for this suggestion. Ubiquitin modification is a crucial way to regulate the stability or function of the substrate proteins. Actually, our data show that OTUD5 does not affect the degradation and stability of TAK1 protein but influences TAK1 phosphorylation and activation both *in vitro* and *in vivo* (Figure 5a-5d, 5g, 5j, 5l, 6c, and 7d). Mechanistically, our findings demonstrate that OTUD5 inhibits TAK1 activation by blocking TAK1-TAB2 interaction via deubiquitinating TAK1 at the K158 site.

To better verify this, we further compared the effects of degradation inhibitors and OTUD5 on TAK1 protein stability. As shown in Figure R1 the proteasome inhibitor MG132 and the lysosome inhibitor chloroquine (CQ) inhibited TAK1 degradation and increased TAK1 levels in podocytes, while OTUD5 overexpression showed no effect on TAK1 protein level. Considering that our data already showed that OTUD5 did not affect the stability of TAK1, we did not put Figure R1 in the revised manuscript.

Figure R1

Reviewer: #1-9. In Figure 7, the addition of the change in inflammatory cytokine expression will be more convincing.

Response: Thank you. We have added the mRNA examinations on inflammatory genes (*Il-6*, *Tnfa*) in the new Supplementary Figure 7c-d.

new Supplementary Figure S7c-d

Reviewer #2 (Remarks to the Author):

In their manuscript the authors investigate the role of DUBs in kidney disease. They used RNAseq to identify dysregulated DUBs and then focus on OTUD5. They use two different mouse models (STZ and STZ+HFD) to determine the role of OTUD5 in combination with an inhibitor for TAK1 (a downstream target of OTUD5) and an overexpression approach of OTUD5. To gain mechanistic insights, mutants of OTUD5 and TAK1 based on known relevant sites are introduced in vitro and the interaction of OTUD5 and TAK1 is determined. Overall, the study is well done and provides new insights. The authors need to improve some aspects in the paper:

Response: Thanks for your positive comments.

Reviewer: #2-1. The authors use the term “diabetic nephropathy”. This should be replaced by “diabetic kidney disease”.

Response: Thanks for your suggestion. We revised the term to “diabetic kidney disease (DKD)” throughout the manuscript.

Reviewer: #2-2. Abstract: The authors write that RNA-seq indicated a significantly increased expression of OTUD5. But in the image shown (Fig. 1a) it is downregulated, which is also congruent with the rest of the text.

Response: Thank you. We apologize for this carelessness and mistake. The “increased expression of OTUD5” in the Abstract should be “decreased expression of OTUD5”. We corrected it.

Reviewer: #2-3. In the introduction the authors state that “developing podocyte specific therapies” remains a major challenge. Are any of their molecular targets podocyte specific? Is this issue addressed by their work?

Response: Thank you for your question.

The phrase "developing podocyte specific therapies" is referenced in a paper in *Nature Reviews Nephrology* (PMID: 22045242). As highlighted in this article, while there are several therapies that can be demonstrated to have direct effects on podocytes, such as glucocorticoids, calcineurin inhibitors, and mTOR inhibitors, they are not specifically targeting podocytes. Some podocyte-specific proteins have been found as potential targets for the treatment of diabetic nephropathy only in pre-clinical studies. For example, Liu et al. reported that podocyte-specific Sirt6 knockout exacerbates podocyte injury in diabetic and adriamycin-treated mice (*Nat Commun*, PMID: 28871079); Sun et al. showed that Cldn5 is predominantly expressed on plasma membranes of podocytes and podocyte-specific Cldn5 knockout exacerbates podocyte injury and proteinuria in a diabetic nephropathy mouse model (*Nat Commun*, PMID: 35332151). Nonetheless, we also think that this sentence “developing podocyte-specific therapies” is overstated and may make confusion. Therefore, we have removed this sentence in the Introduction section.

Reviewer: #2-4. The phenotype in podocyte specific ko mice is minor. There is only a minor increase of serum creatinine, serum BUN and urine ACR.

Response: Thank you. In our study, all mice were fed in hyperglycemia for 16 weeks to induce nephropathy. We may think the basal DN phenotypes in OTUD5^{f/f}(WT) diabetic mice are very serious, therefore, the further increased phenotypes in some biomarkers by OTUD5CKO look minor. However, the differences between OTUD5^{f/f}-DM group and OTUD5CKO-DM group are

statistically significant. As shown in Figure 2 and 3, OTUD5CKO significantly exacerbated the DN phenotypes in both T2DM and T1DM mouse models.

Reviewer: #2-5. The observed changes in regard to the inflammatory cytokines IL6 and Tnfa are more pronounced. Does this translate into a different composition of immune cells in the kidney? Are cytokines in the urine affected?

Response: Thank you for your question. Accordingly, we tried to examine the immune cell infiltration in the diabetic kidney using immunohistochemical staining for macrophage marker CD68. We found that OTUD5CKO mice exhibits more macrophage infiltration in T2DM mouse kidney. Moreover, we observed increased mRNA expression of chemokines (*Ccl2* and *Cxcl10*), which are responsible to macrophage infiltration, in OTUD5CKO-T2DM mice compared to the OTUD5^{fl/fl}-T2DM mice. We think that the OTUD5 deficiency in podocytes also further increased the chemokine levels in renal tissues of T2DM mice, leading to more macrophage infiltration in kidneys. These results are shown in the new Supplementary Figure S2d-f.

In addition, we used ELISA to detect the expression of inflammatory cytokines in the urine. The results showed that there were no significant changes in the expression of IL-6 and TNF- α in urine (Figure R2). Since most papers on DKD did not report the urine cytokine levels, we did not put Figure R2 in the revised manuscript.

new Supplementary Figure S2d-f

Figure R2

Reviewer: #2-6. Fig. 6: Takinib has a strong basal phenotype. This makes it difficult to draw a conclusion.

Response: Thank you. This concern is same to the Reviewer #2-15. Please see our response to Reviewer #2-15.

Reviewer: #2-7. More methodological information on how MPC5 cells were cultured and differentiated are needed. In particular how were cells stimulated with glucose and PA? concentration, duration, change of medium etc. Details are missing.

Response: Thank you for your comments. We added these required information in the revised manuscript.

Specifically, the mice renal podocyte cell line (MPC5) was cultured and differentiated as previously described (*Kidney Int*, PMID:17457377). In brief, MPC5 cells were cultured in RPMI 1640, 10% fetal bovine serum (Meilunbio, PWL001) and 10 IU/mL of recombinant murine c-interferon (IFN- γ , Invitrogen, CA, USA) at 33 °C and 5% CO₂. To induce differentiation, the cells were transferred to 37 °C for 10-14 days and the medium was replaced without IFN- γ . After differentiation, MPC5 cells were stimulated with 33 mM glucose and 200 μ M PA for indicated times in respective experiments.

Reviewer: #2-8. Were mice on a C57BL/6 “j” or “n” substrain used. Please provide this information.

Response: Thanks. In this study, we used C57BL/6J substrain and C57BL/6J background mice. This information was added in the revised Methods.

Reviewer: #2-9. In the HFD/STZ model: please state clearly in the methods how often STZ was injected. What was the bodyweight in these mice?

Response: Thank you. Both OTUD5^{fl/fl}-Ctrl group and OTUD5CKO-Ctrl group were given a normal diet, while the OTUD5^{fl/fl}-T2DM group and OTUD5CKO-T2DM group were given a high-fat diet for 4 weeks and then administered with a single dose of STZ (100 mg/kg) via i.p. injection to generate the T2DM model (*Kidney Int*, PMID: 30791996; *Redox Biol*, PMID: 37597421). “A single dose of STZ (100 mg/kg)” has been added to the revised manuscript.

In addition, we have showed the body weight data of mice in different groups in the new supplementary Figure S2c.

new Supplementary Figure S2c

Reviewer: #2-10. Statistics: Was correction for multiple testing performed? Were data normally distributed? How as this ascertained?

Response: Thank you. We performed correction for multiple testing in our analysis. One-way ANOVA analysis of variance was used for comparing multiple groups in GraphPad Prism 8.0 software. In the process of analysis, we selected the recommended "corr for multiple comparisons using statistical hypothesis testing". In addition, the data is normally distributed. For the animal experiment data, the Shapiro-Wilk test yielded a P-value greater than 0.05, indicating that the data were normally distributed. For the cell experiment data, we acquiesced that the data followed a normal distribution due to the large number of cells involved in each experiment. We described the statistical analysis in details and we noted the respective statistical analysis method in all figure legends in the revised manuscript.

Reviewer: #2-11. Fig. S2A and corresponding text: The authors state that “deletion of OTUD5 in

podocytes was validated by tail genotyping”. That is not possible. There is something obviously wrong.

Response: Thank you. We have revised the corresponding text and Figure S2a in the revised manuscript. We generated podocyte-specific Otud5 knockout mice (conditional knockout, OUTD5CKO) by crossing Otud5^{fl/fl} mice and Nphs1-iCre mice, which were identified by tail genotyping.

Updated Supplementary Figure S2a

Reviewer: #2-12. In Fig. 1E the authors show a reduction of OTUD5 in the T2DM model, which is not apparent Fig. 4D.

Response: Thank you. In the original Fig. 4d, the OTUD5 level (in input) in T2DM group is lower than that in the control group, while it is not as apparent as that in Fig. 1e. The reason for this may be the relatively low protein concentration in input lysate. To address this issue, we have re-performed this experiment using increased protein concentrations in input lysate and replaced the more representative images in the updated Fig. 4d.

Updated Figure 4d

Reviewer: #2-13. Can the authors explain why in Fig. 4J ubiquitination remains high in the K158R mutant? If this site is required for de-ubiquitination, as stated by the authors, I would expect less ubiquitination at baseline. Instead, there is no reduction in ubiquitination. How do the authors explain this observation?

Response: Thank you. In Figure 4j, we observed that the K158R mutation abolished OTUD5-mediated deubiquitination, suggesting that lysine residue K158 of TAK1 is critical for OTUD5 deubiquitination.

As commented by the reviewer, it is interesting that there is no reduction in ubiquitination level of K158R mutant compared to the wide-type TAK1. We try to explain this observation as following. It is worth noting that ubiquitination pathways are complex and involve multiple ubiquitin ligases and deubiquitinating enzymes. The ubiquitination/deubiquitination happens at many lysine sites in TAK1. It is possible that, when K158 is mutated and can't be ubiquitinated, the ubiquitination of TAK1 at other lysine sites is increased to compensate for the loss of ubiquitination at the K158 site. This kind of phenomenon on TAK1 ubiquitination has been reported previously. For example, Wang et al. reported that K282 and K547 were specific sites for TRIM16-catalyzed ubiquitination of TAK1, while K282R and K547R mutants of TAK1 also showed ubiquitination levels at the baseline of wide-type TAK1 (*Cell Metab*, PMID: 34146477).

Reviewer: #2-14. Fig. 5A: based on one immunoblot not statement regarding “significance” is possible. Please show multiple blots and quantification (e.g. in the supplements). Quantification is also needed for Fig. 5E,F.

Response: Thank you. We have provided quantification data for Fig. 5a, 5e, and 5f. The data are shown in the new Supplementary Figure 5a, 5d, and 5e, respectively.

new Supplementary Figure S5a,5d,5e

Reviewer: #2-15. Fig. 6: Takinib has a strong basal phenotype (compare OTUD5^{f/f}-T2DM versus OTUD5^{f/f}-T2DM+Takinib. This may be a dominant effect, which makes it difficult to draw a conclusion. Please comment on this.

Response: Thank you. TAK1 is a well-known kinase regulating inflammatory signaling pathway and previous studies have demonstrated the pivotal regulatory role of TAK1 in podocyte inflammation and DN progression. Pharmacological inhibitor of TAK1 significantly reduce the expression of urinary albumin, histological changes and renal inflammatory cytokines induced by DKD (*Int Immunopharmacol*, PMID: 27268284). Here, we found that podocyte OTUD5 alleviates DKD through deubiquitinating TAK1 and preventing TAK1 activation. In Figure 6, when we treated the T2DM mice with TAK1 inhibitor Takinib, the phenotypes of DKD were significantly normalized and OTUD5 knockout could not further induce the DKD pathology in Takinib-treated mice. These data further highlight the potential of TAK1 as therapeutic target of DKD and validate the efficiency of TAK1 inhibitor in treating DKD. However, since Takinib inhibited the whole-body TAK1, it showed a strong basal phenotype in OTUD5^{f/f}-T2DM mice, which may make it very difficult for podocyte OTUD5 deficiency to aggravate the DKD phenotypes in Takinib-treated mice. It may be a limitation of this study. The use of podocyte-specific TAK1 inhibition/knockout may be preferable to investigate the podocyte OTUD5-TAK1 axis in podocytes in DKD. We discussed this point and acknowledged this limitation in the revised Discussion.

Reviewer: #2-16. Fig. S7A: please show markers for podocytes and other cells and also analyze other cells to demonstrate that indeed podocyte specific overexpression was achieved. This is important as the authors emphasize this aspect, e.g. in the discussion.

Response: Thank you. According to this suggestion, we added to perform an immunofluorescence double-staining using the kidney tissue sections of OTUD5^{f/f}-T2DM-AAV-OTUD5 mice. The results in the new Supplementary Figure 7b show that OTUD5 is mainly expressed in the Nephryn-positive podocytes, rather than in Desmin-positive mesangial cells and AQP1-positive tubular

epithelial cells, indicating that the podocyte specific OTUD5 overexpression was achieved by the constructed AAV system.

new Supplementary Figure S7b

Reviewer: #2-17. A major limitation is the lack of human data. Also, no mouse model independent of STZ was used to validate their observations. At least basal phenotypes can be shown in humans and alternative mouse models.

Response: Thank you for this suggestion. Accordingly, we collected renal biopsies from 3 diabetic subjects and 3 nondiabetic control subjects and examined the OTUD5 levels in these biopsies by immunofluorescence staining. As shown in the new Figure 1e, the level of OTUD5 expression in renal biopsies from DKD subjects is much lower than that in non-DKD subjects.

In addition, we conducted western blot and qPCR analyses for OTUD5 expression on the kidney tissues from NOD and db/db diabetic mice, two well-established mouse models of spontaneous diabetes independent of STZ. We confirmed the down-regulated protein and mRNA levels of OTUD5 in both NOD and db/db mice compared with control mice. These results are shown in the new Figure 1h-i and new Supplementary Figure S1d.

new Figure 1e

new Figure 1h-i

new Supplementary Figure S1d

Reviewer: #2-18. The term “significant” should only be used in the context of statistical analyses; e.g. line 69 - replace “significance of podocyte injury”.

Response: Thank you. We have replaced "significance" with "importance" in the sentence.

REVIEWERS' COMMENTS

Reviewer #1 (Remarks to the Author):

All issues are addressed. No more comments.

Reviewer #2 (Remarks to the Author):

Thanks for the revised version of the manuscript. The authors did a great job in addressing the points, including adding human data to their publication, which adds translational relevance. I have only two remaining comments:

Ad 2.4: The difference in albuminuria, Cre, and BUN is small. Based on the Brosius, JASN 2009 consensus paper, the increase of albuminuria in murine models of diabetes should be 10-fold and the decrease of Creatinine clearance should be 50%. This is agreeable rarely observed and hence I do not want to overemphasize this. Yet, in the context of DKD, the primary endpoints should be albuminuria or Creatinine – and both increase less than 1.5-fold. This does not exclude a pathogenic function of the proposed mechanism, but the proposed mechanism is probably interacting with others. The authors should acknowledge this in their discussion.

Ad 2-13: The authors provide an explanation for the observation made. I would suggest to briefly allude to this issue in the discussion, as other readers may have the same problem.

Responses to the Reviewers' comments

Reviewer #1 (Remarks to the Author):

All issues are addressed. No more comments.

Reviewer #2 (Remarks to the Author):

Thanks for the revised version of the manuscript. The authors did a great job in addressing the points, including adding human data to their publication, which adds translational relevance. I have only two remaining comments:

Response: Thanks for your positive comments.

Reviewer: #2-1. Ad 2.4: The difference in albuminuria, Cre, and BUN is small. Based on the Brosius, JASN 2009 consensus paper, the increase of albuminuria in murine models of diabetes should be 10-fold and the decrease of Creatinine clearance should be 50%. This is agreeable rarely observed and hence I do not want to overemphasize this. Yet, in the context of DKD, the primary endpoints should be albuminuria or Creatinine - and both increase less than 1.5-fold. This does not exclude a pathogenic function of the proposed mechanism, but the proposed mechanism is probably interacting with others. The authors should acknowledge this in their discussion.

Response: Thank you. We acknowledge that the alterations in albuminuria, creatinine, and BUN in our diabetic model did not reach the degree as large as that in the 2009 JASN paper by Brosius. In our study, we mainly focus on the podocyte loss, which is an early event in DKD development. Therefore, we selected a relatively short 4-month hyperglycemia modeling to facilitate the observation of podocyte loss. This shorter duration of hyperglycemia may account for the relatively weak changes in biochemical indicators of DKD. Some previous studies have also reported similar changing folds in these indicators using the DKD mice (*Kidney Int*, PMID: 32739204; *Int J Biol Sci*, PMID: 37928264). We do think that it would be more compelling to further validate the role of OTUD5 in a DKD model with longer hyperglycemia duration and more severe kidney damage. We discussed this limitation in the revised Discussion.

Reviewer: #2-2. Ad 2-13: The authors provide an explanation for the observation made. I would suggest to briefly allude to this issue in the discussion, as other readers may have the same problem.

Response: Thank you. We discussed this point in the revised Discussion. It is worth noting that there is no reduction in ubiquitination level of K158R mutant compared to that of wide-type TAK1. As we know, the ubiquitination/deubiquitination happens at many lysine sites in TAK1. It is possible that the ubiquitination at K158 accounts for only a small portion in the total TAK1 ubiquitination or the ubiquitination of TAK1 at other lysine sites is increased to compensate for the loss of ubiquitination at the K158 site.